# Vinci: Deep Thinking in Text-to-Image Generation using Unified Model with Reinforcement Learning

**Wang Lin**[1,*]**, Wentao Hu**[2,*]**, Liyu Jia**[2]**, Kaihang Pan**[1]**, Majun Zhang**[1]**,**
**Zhou Zhao**[1]**, Fei Wu**[1]**,Jingyuan Chen**[1,†]**, Hanwang Zhang**[2]
[1]Zhejiang University, [2]Nanyang Technological University
linwanglw@zju.edu.cn,  jingyuanchen@zju.edu.cn

## Abstract

With the continuous development of large language models and reasoning chain technologies, the potential of deep reasoning based on reinforcement learning has shown remarkable promise in multi-task scenarios. However, existing unified models have yet to achieve end-to-end integration in image generation and understanding tasks, limiting the model's self-reflection ability and the realization of cross-modal reasoning chains. To address this, we propose Vinci, a novel framework designed to enable interleaved image generation and understanding through deep reasoning capabilities. We leverage a small amount of multimodal chain-of-thought (MCoT) data for cold-start and employ reinforcement learning to guide the integration of image generation and understanding tasks. Additionally, we introduce a momentum-based reward function, which dynamically adjusts the reward distribution by considering historical improvements, ensuring the stability of the model across multiple generations. Experimental results demonstrate that integrating MCoT can achieve a +22% improvement over the base model on Geneval, effectively enhancing both image generation quality and instruction alignment capabilities.

## 1 Introduction

With the rapid development of large language models [1, 9, 82, 61] and Chain of Thought [49, 37, 75, 32] (CoT) techniques, deep thinking through reinforcement learning has become a key research focus in artificial intelligence. It has shown exceptional reasoning capabilities in tasks like mathematical problem-solving [62, 18] and code generation [29, 28, 61], significantly improving both model performance and interpretability by providing clear, step-by-step deduction and decision-making.

This step-by-step reasoning ability is being applied to cross-modal tasks, particularly in image generation and understanding. In image understanding, many structured reasoning mechanisms [85, 79] have been proposed to enhance controllability and interpretability. In image generation, [10, 14, 57] focuses on refining the input prompts or employing text-based layout construction prior to synthesis. Recently, breakthroughs in unified models have enabled a single model to simultaneously perform image generation and understanding. However, the potential of Chain of Thought techniques in existing unified models has yet to be fully explored.

The main challenge lies in the lack of end-to-end integration between image generation and understanding. For example, in models like Janus-Pro [5] and Show-o [78], images are represented in two ways within the model (*i.e.*, the model cannot directly understand the representations of the images it generates), thus limiting the model's ability to reflect on and optimize the generated results. Furthermore, while there have been attempts [89, 31, 22] to introduce CoT into these models, they are mostly limited to unimodal reasoning chains. For instance, in the T2I-R1 [31] model, semantic-level CoT is applied only to enhance text prompts, while token-level CoT is optimized only on image tokens.

39th Conference on Neural Information Processing Systems (NeurIPS 2025).

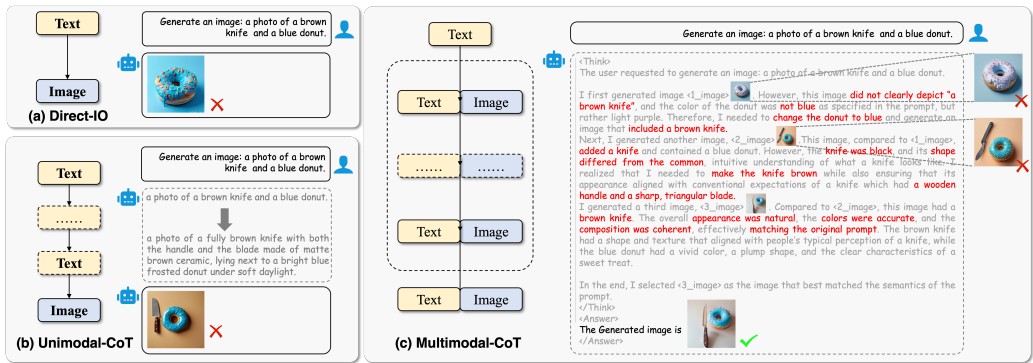

Figure 1: The illustration of CoT in text-to-image generation, showing three different approaches: (a) Direct-IO, where the image is generated directly from the prompt, (b) Unimodal-CoT, which involves generating an extended prompt before producing the image, and (c) Multimodal-CoT, where both text and image are processed iteratively to refine the image generation process based on the evolving understanding of the prompt.

This unimodal approach to reasoning chains restricts the model's ability to perform cross-modal deep thinking, further limiting its performance improvements.

Our goal is to integrate interleaved Multimodal Chain-of-Thought (MCoT) capabilities into the model, allowing it to observe and reflect on its generated results in real-time, thus achieving deep integration of image generation and understanding. As shown in Figure 1, initially, the model generates an image from the prompt *"a brown knife and a blue donut"*. Upon reviewing the first image, the model identifies that the donut is not blue and the knife is missing. It reflects on these issues and adjusts the generation strategy to correct the donut's color to blue and adds a knife, but the knife's color and shape still do not match typical expectations. Realizing the mistake, the model gives the knife's appearance description. This iterative process reflects the deep thinking of image generation and understanding, refining the generated images based on real-time evaluation and adjustments.

To achieve this, this paper introduces **Vinci**, which is the first image generation model capable of deep thinking. Specifically, we employ a progressive training strategy and fine-tune the model using a small amount of MCoT data to provide initial CoT guidance. Subsequently, we use reinforcement learning to guide the model to simultaneously focus on both image generation and image understanding tasks during the generation process. In addition to applying the reward model to evaluate the generated images, we introduce a momentum reward function to further enhance the model's stability and generation quality. The core of this method lies in dynamically distributing process rewards by considering the improvements made by the model in historical generation processes. By employing this approach, the Vinci can better balance the model's immediate performance with its long-term stability, preventing performance fluctuations or degradation over multiple iterations. Our contributions are summarized as follows:

- We propose **Vinci**, the first image generation model capable of deep reasoning, which integrates interleaved MCoT capabilities. This enables the model to not only generate images but also observe and reflect on its outputs in real-time.

- We propose a momentum-based reward function that dynamically distributes process rewards based on historical improvements. This approach enhances the model's stability while balancing immediate performance with long-term stability.

- Experimental results demonstrate that integrating MCoT can achieve a +22% improvement over the base model on Geneval [17] benchmarks.

## 2 Related Work

### 2.1 Unified Generation and Understanding LMM

Recent advancements in multimodal understanding and image generation have led to efforts to integrate these tasks into a unified framework. The main approaches can be categorized into two

types. (1) Fused autoregressive and diffusion models such as Transfusion[88], Show-o[78], and MonoFormer [86], combine the strengths of autoregressive models for text generation with the high-quality image generation of diffusion models. Text is generated autoregressively, while images are produced through a multi-step denoising process. It balances symbolic control with visual fidelity, but fails to perform generation and understanding simultaneously. (2) Autoregressive models unify image and text generation within a sequential framework, using powerful autoregressive language models (e.g., LLaMA [1], Vicuna [9]) at their core. These models encode images into discrete visual tokens and process them alongside text tokens. Various image tokenization strategies, including pixel-based like LWM [50], Chameleon [65], and ANOLE [8], semantic-based like Emu [64], LaVIT [35], and DreamLLM [12], and learnable query encoding like SEED [15] and MetaQueries [55], allow the integration of visual information within the autoregressive framework. These autoregressive models demonstrate significant improvements in both image generation quality and efficiency. Note, we aim to perform generation and understanding simultaneously. Thus, the autoregressive models such as Chameleon [65] and Emu3 [71], which have a unified image representation, provide the foundation for MCoT in image generation.

## 2.2 Multimodal-CoT Reasoning

Multimodal Chain-of-Thought (MCoT) has gained significant attention in tasks such as Visual Question Answering (VQA), where early works like IPVR [6] and Multimodal-CoT [85] laid the foundation by generating intermediate rationales before final predictions. Subsequent research improved MCoT by introducing self-consistency [72] with word-level majority voting and dynamic reasoning selection based on human cognitive strategies. Frameworks like CoCoT [84] and RelationLMM [77] enhanced multi-image comprehension and object relationship modeling. Methods such as DDCoT [87] and Socratic Questioning [25] employed staged reasoning for better interpretability, while Chain-of-Spot [51] and DCoT [30] focused on region-of-interest analysis to improve contextual understanding. Recent advancements expanded MCoT beyond VQA. Techniques like G-CoT [52] and STIC [11] addressed data annotation limitations through automated augmentation, and DPMM-CoT [24] regenerated image features from the latent space to handle complex reasoning. Additionally, multimodal rationales, as seen in Visual-CoT [59] and MVoT [39], have enhanced reasoning across modalities, improving overall model performance. In image generation tasks, models like Prompt-CoT [83] and LayoutLLM-T2I [57] optimize input prompts and layout construction, respectively, to improve image synthesis quality. These developments highlight the growing importance of MCoT in multimodal reasoning, enabling models to tackle complex image-related tasks with greater efficiency and interpretability.

## 3 Method

### 3.1 Preliminary

Recently, reinforcement learning has become the primary method for unlocking the reasoning capabilities of large language models (LLMs). [62] introduces the Group Relative Policy Optimization (GRPO) framework. Unlike reinforcement learning algorithms such as PPO [60] that require a critic model to evaluate policy performance, GRPO compares groups of candidate responses directly, eliminating the need for an additional critic model. Given the input instruction $q$, GRPO first generates $G$ distinct predictions $\{o_1, o_2, ..., o_g\}$ from the old policy $\pi_{\theta_{old}}$. Then, GRPO takes actions based on these predictions and denotes the obtained rewards as $\{r_1, r_2, ..., r_g\}$. By computing their mean and standard deviation for normalization, GRPO determines the relative quality of these responses:

$$A_i = \frac{r_i - \text{mean}(\{r_1, \ldots, r_g\})}{\text{std}(\{r_1, \ldots, r_g\})}, \tag{1}$$

where $A_i$ represents the relative quality of the $i$-th answer. The GRPO method employs a clipped objective function, similar to PPO, and introduces a KL penalty term that compares the current policy $\pi_\theta$ with the reference model $\pi_{\theta_{\text{ref}}}$ into the loss, as follows:

$$\mathcal{L}_{\text{GRPO}}(\theta) = -\frac{1}{G} \sum_{i=1}^{G} \frac{1}{|o_i|} \sum_{t=1}^{|o_i|} \tag{2}$$

$$\left[ \min\left( r_{i,t}(\theta)\hat{A}_{i,t}, \text{clip}\left( r_{i,t}(\theta), 1 - \epsilon, 1 + \epsilon \right) \hat{A}_{i,t} \right) - \beta \mathbb{D}_{KL} \left[ \pi_\theta || \pi_{\text{ref}} \right] \right] \tag{3}$$

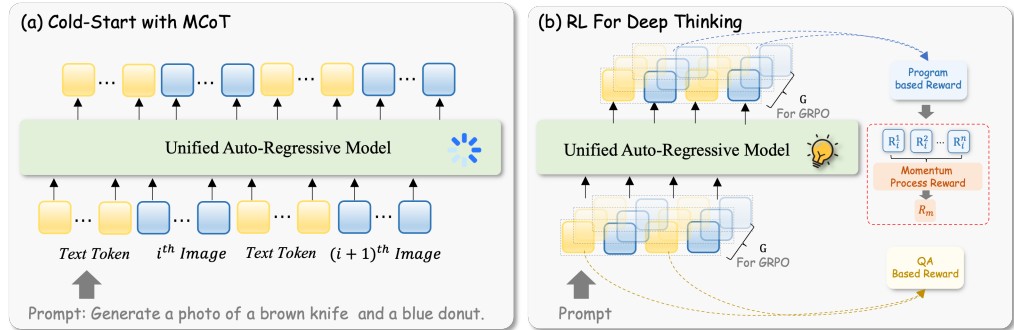

Figure 2: Overview of the Vinci model architecture. (a) Cold-start for long Multimodal Chain-of-Thought (MCoT), where the unified autoregressive model sequentially processes text tokens and image tokens. (b) Reinforcement Learning (RL) for deep thinking, where the model incorporates multiple reward signals (program-based, momentum process, and QA-based rewards) to iteratively refine the generated content during the generation process.

where $r_{i,t}(\theta)$ represents the ratio between the probabilities of $\pi_\theta$ and $\pi_{\theta_{\text{old}}}$ for generating the current token:

$$r_{i,t}(\theta) = \frac{\pi_\theta(o_{i,t}|q, o_{i,<t})}{\pi_{\theta_{\text{old}}}(o_{i,t}|q, o_{i,<t})}. \tag{4}$$

In text-to-image generation, the model is guided to follow a predefined template to produce the reasoning process and final images. The reward functions evaluate the generated image in visual RL and provide feedback for optimization.

## 3.2 Cold Start with Multimodal-CoT

Inspired by recent advances in reasoning models [19, 61, 53], we aim to incorporate a long-chain-of-thought reasoning into the unified models generation process as shown in Figure 2. However, existing unified models are typically designed to generate images in a single forward pass, lacking the ability to perform multi-round understanding and reflection in the image generation process. We first introduce a cold-start stage to adjust the model's output format and prevent instability of RL training during early-stage.

**Multimodal-CoT Data Collection.** Unlike Unimodal-CoT, which only refines the prompt without considering the generated image, the Multimodal-CoT we introduce refers to a process in which the model engages in understanding and reflecting on intermediate images during text-to-image generation. In each generation attempt, the model analyzes the generated image, identifies inconsistencies or areas for improvement, and proposes adjustments to better align with the prompt and enhance visual quality. This iterative, interleaved process of textual reasoning and visual feedback ultimately leads to the generation of the final, high-quality image.

To construct Multimodal-CoT data for the cold-start stage, we design a three-stage data pipeline.

- **Generate the Image for MCoT.** For each text query $q$, we generate a set of $n$ candidate images, denoted as $\{p_1, p_2, \ldots, p_n\}$. We then apply an object detection model to each image to identify objects and obtain detection outputs, resulting in a set of tuples $\{(p_1, o_1), (p_2, o_2), \ldots, (p_n, o_n)\}$, where $o_i$ represents the detected objects in image $p_i$.

- **Generate the Text for MCoT.** To prevent confusion in multi-image understanding, we first require the multimodal large model (MLLM) to independently evaluate each generated tuple $(p_i, o_i)$ for its semantic alignment with the original query and overall quality. Note that we also provide the output of the object detection model $o_i$, which helps reduce hallucinations in the multimodal large model. In this process, MLLM generates a caption $c_i$ and a score $g_i$ for each image $p_i$. These outputs are then combined into annotated triples $(p_i, c_i, g_i)$, which are sorted by their scores in ascending order, resulting in the final set $\{(p_1, c_1, g_1), (p_2, c_2, g_2), \ldots, (p_n, c_n, g_n)\}$, where $g_1 \leq g_2 \leq \ldots \leq g_n$.

- **MCoT Construction:** Given a predefined reasoning step count $k$, we randomly select $k$ triples from this sorted set for combination. It is important to maintain the relative order

of the triples according to their scores and ensure that each combination contains at least one high-scoring (correct) triple, which serves as the endpoint for the MCoT. This results in combinations of the form $\{(p_i^1, c_i^1, g_i^1), \ldots, (p_i^k, c_i^k, g_i^k)\}$, where $k \leq n$. Each combination is then input into the MLLM to construct the final MCoT data.

For the final constructed MCoT, we use GPT-4o [53] to evaluate and filter out data that contains hallucinations or failures at the MCoT endpoints. Through these three stages, we construct an interleaved multimodal sequence that represents an iterative process of generation and reasoning.

### 3.3 Reinforcement Learning For Deep Thinking

**Reward Functions Based on Comprehension Models** The overall design philosophy of our reward model is to utilize comprehension models to evaluate both the generated image and text in RL. In this paper, we categorize the comprehension-based reward into two major types:

*Program-based Reward Function.* For the generated image, the comprehension model should understand and evaluate the consistency between the generated image and the textual prompt. Given the prompt like "*3 clocks and 1 dog*" and generated image, we can use visual detectors [4] to evaluate the generation quality. For example, we count the clocks based on the detector's confidence, returning 1 if the count is correct and 0 otherwise. Each prompt has its own item sets to be tested, and the average of the scores for each test is used as the reward score $R_i$.

*QA-based Reward Function for Generated Text.* For the generated text, the comprehension model should evaluate whether the text accurately describes the generated image. Given both the generated image and text, like in Figure 1 (c), we leverage the Multi-modal Large Language Models (MLLMs) to judge the quality of the evaluation text for the generated image. We require the model to be evaluated on three aspects: the completeness of the image description, whether it identifies issues in the image, and whether it includes strategies for improvement. Each aspect is rated on a scale of 0, 1, or 2, with the final reward score $R_t$ being the average of the three scales.

**Momentum Process Reward Function** Given a sequence of $n$ generated images $\{I_t\}_{t=1}^n$, let $s_t \in [0, 1]$ denote the quality score of the $t$-th image produced by the reward function. We aim to design a process reward function $R_m$ that simultaneously evaluates both instantaneous quality and improvement dynamics.

Inspired by the Adam optimizer [36], we propose a dynamic process reward that is used to encourage consistent improvements as follows:

$$R_m = \frac{1}{n} \sum_{t=1}^n R_i(s_t) + \alpha \sum_{k=1}^{n-1} \frac{\Delta_k}{\sqrt{V_k} + \epsilon} \lambda^{k-1} \tag{5}$$

where the hyperparameter $\alpha$ balances between absolute quality and improvement momentum, the time decay factor $\lambda \in (0, 1)$ imposes attenuation on later improvements, and $\epsilon = 10^{-8}$ ensures numerical stability.

The current improvement $\Delta_k = \max(0, R_i(s_k) - R_i(s_{k-1}))$, quantifies non-negative increments between generated images. With the momentum normalization term $\Delta_k/\sqrt{V_k}$, which establishes dynamic scaling based on historical improvement patterns. This reward function achieves three objectives. First, it normalizes current improvements against historical volatility, preventing disproportionate rewards from isolated quality spikes. Second, the exponentially weighted variance calculation prioritizes recent trends while maintaining memory of long-term patterns. Third, the nonlinear response curve generates superlinear rewards when sustained improvements exceed historical baselines ($\Delta_k \gg \sqrt{V_k}$), while penalizing inconsistent progress through variance accumulation.

The historical volatility $V_k = \gamma V_{k-1} + (1 - \gamma)\Delta_k^2$, evolves through exponential moving averaging. The momentum decay rate $\gamma \in [0, 1)$ controls historical variance adaptation. This formulation ensures older improvements contribute diminishing weights to the variance estimate, creating adaptive resistance to quality fluctuations.

The $\lambda^{k-1}$ term implements exponential temporal decay to prioritize early-stage improvements, modeling the empirical observation that establishing baseline quality early enables more effective subsequent refinements. The decay schedule follows $\lambda^{k-1} = e^{(k-1)\ln \lambda}$, which creates implicit temporal milestones where early improvements receive more weighting, while late optimizations

| Type | Method | Single Obj. | Two Obj. | Counting | Colors | Position | Color Attr. | Overall |
|------|--------|-------------|----------|----------|--------|----------|-------------|---------|
| | PixArt-$\alpha$ [3] | 0.98 | 0.50 | 0.44 | 0.80 | 0.8 | 0.70 | 0.48 |
| | SDXL [56] | 0.98 | 0.74 | 0.39 | 0.85 | 0.15 | 0.23 | 0.55 |
| *Gen. Only* | FLUX.1-dev [38] | 0.98 | 0.79 | 0.73 | 0.77 | 0.22 | 0.45 | 0.66 |
| | DALL-E 3 [63] | 0.96 | 0.87 | 0.47 | 0.83 | 0.43 | 0.45 | 0.67 |
| | CogView4-6B [2] | 0.99 | 0.86 | 0.66 | 0.79 | 0.48 | 0.58 | 0.73 |
| | SD3-Medium [13] | 0.99 | 0.94 | 0.72 | 0.89 | 0.33 | 0.60 | 0.74 |
| | SEED-X [16] | 0.97 | 0.58 | 0.26 | 0.80 | 0.19 | 0.14 | 0.49 |
| | Emu3-Gen [71] | 0.98 | 0.71 | 0.34 | 0.81 | 0.17 | 0.21 | 0.54 |
| | TokenFlow-XL [58] | 0.95 | 0.60 | 0.41 | 0.81 | 0.16 | 0.24 | 0.55 |
| | Transfusion [88] | - | - | - | - | - | - | 0.63 |
| | D-DiT [41] | 0.97 | 0.80 | 0.54 | 0.76 | 0.32 | 0.50 | 0.65 |
| *Unified.* | Show-o [78] | 0.98 | 0.80 | 0.66 | 0.84 | 0.31 | 0.50 | 0.68 |
| | ILLUME+ [26] | 0.99 | 0.88 | 0.62 | 0.84 | 0.42 | 0.53 | 0.72 |
| | Infinity [23] | - | 0.85 | - | - | 0.49 | 0.57 | 0.73 |
| | Janus-Pro-7B [5] | 0.99 | 0.89 | 0.59 | 0.90 | 0.79 | 0.66 | 0.80 |
| | GPT-4o [53] | 0.99 | 0.92 | 0.85 | 0.91 | 0.75 | 0.66 | 0.85 |
| | Vinci(Ours) | 0.99 | 0.86 | 0.48 | 0.83 | 0.86 | 0.54 | 0.76 |

Table 1: Evaluation of text-to-image generation ability on GenEval benchmark. Emu3-Gen is our base model. *Gen. Only* indicate models that can only do generation tasks, *Unified.* indicate models that can do both generation and understanding tasks

contribute minimally to the momentum score. The decay rate $\ln \lambda$ controls the transition steepness between these regimes.

To effectively guide the model's learning process and ensure that it can generate high-quality outputs, we combine the three reward functions as:

$$R = R_i + R_t + R_m \tag{6}$$

By integrating these three components, the overall reward function leads to more accurate, coherent, and consistent multimodal generation.

## 4 Experiments

### 4.1 Experimental Setup

**Training Data.** Our training data consists of two distinct phases. In the cold-start phase, we randomly generated 20,000 MCoT; the images are generated by Flux [38] and the text is generated by Qwen-VL [68]. Following this, the reinforcement learning (RL) phase began with the utilization of 30,000 prompts, all without corresponding images. Notably, to ensure the integrity of the training and evaluation processes, we performed careful deduplication of the training prompts, eliminating any overlap with the test set. This measure was critical in preventing data leakage and ensuring a robust evaluation of the model's performance on unseen prompts.

**Implementation details.** For our training, we adopted Emu-Gen as the base model, which has a unified image representation. The model's context length was extended to 15,360 tokens, which allowed for processing longer sequences of text and image pairs and supported approximately three iteration within each context window. The learning rate was set to 1e-5 and a beta of 0.01.

### 4.2 Main Results

The experimental results presented in Table 1 highlight the performance of various models on the GenEval [17] benchmark, which evaluates the text-to-image generation ability across different tasks. The results highlight the superior performance of Vinci over the *Gen. Only* models, demonstrating the necessity of integrating comprehension abilities into generative models for enhanced instruction following capabilities.

In comparison with the *Unified* models, Vinci only lags behind Janus-Pro-7B [5] and GPT-4o [53], indicating its strong competitiveness among models capable of both generation and understanding tasks. The enhancements in Vinci over the baseline model Emu3-Gen [71] are particularly significant across several dimensions. Specifically, Vinci elevates the Position generation score from 0.17 to 0.86

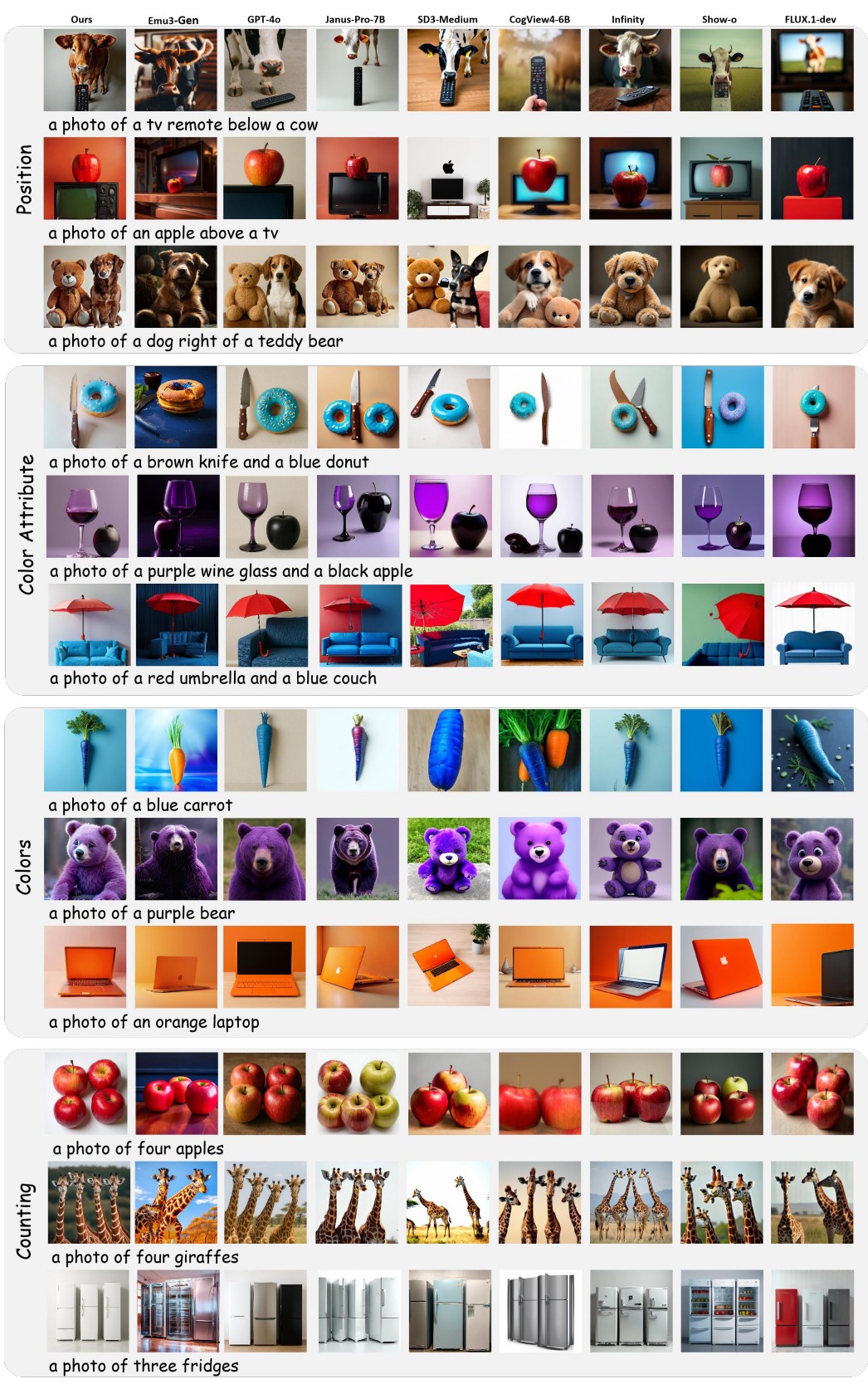

Figure 3: Qualitative experimental results of Vinci. Compared to existing text-to-image generation models, the images generated by Vinci demonstrate better alignment with the given prompts.

| Method | Single Obj. | Two Obj. | Counting | Colors | Position | Color Attr. | Overall |
|---|---|---|---|---|---|---|---|
| Emu3-Gen [71] | 0.98 | 0.71 | 0.34 | 0.81 | 0.17 | 0.21 | 0.54 |
| $w.R_t$ | 0.98 | 0.72 | 0.31 | 0.77 | 0.51 | 0.30 | 0.59 |
| $w.R_i$ | 0.96 | 0.73 | 0.36 | 0.75 | 0.72 | 0.44 | 0.66 |
| $w.R_i\&R_t$ | 0.99 | 0.84 | 0.44 | 0.79 | 0.75 | 0.46 | 0.71 |
| $w.R_i\&R_t\&R_m$ | 0.99 | 0.86 | 0.48 | 0.83 | 0.86 | 0.54 | 0.76 |

Table 2: Ablation study results on the GenEval benchmark. We compare the performance of the base model Emu3-Gen and its variants with modified reward functions.

(+0.69) and Color attribution score from 0.21 to 0.54 (+0.33). These improvements underscore the efficacy of the Multimodal-CoT in achieving a deep integration of image generation and understanding. By enabling the model to observe and reflect on its generated results in real-time, Vinci not only generates images but also enhances the interpretability and controllability of the generated content.

Furthermore, the results in Figure 3 show that Vinci demonstrates a robust performance across a variety of tasks. For instance, when tasked with generating an image of a "tv remote below a cow" or an "apple above a tv" Vinci successfully positions the objects as per the instructions, showcasing its ability to understand and execute complex instructions involving multiple objects.

## 4.3 Ablation Study

The ablation study results presented in Table 2 provide valuable insights into the impact of different reward functions on the performance of the Vinci model.

**Image understanding ability can feed back into image generation.** Initially, when the model is trained using only the text quality evaluation through $R_t$, we observe improvements in image generation, particularly in areas such as position and color attribution. This result suggests that improving the model's ability to understand the generated image can have a positive effect on the image generation process itself.

**Explicit image quality evaluation is important.** By providing explicit feedback on the image quality, $R_i$ acts as a direct optimization signal, guiding the model to produce higher-quality images. The explicit evaluation of image fidelity ensures that the generated images are not only contextually accurate but also visually compelling. The model is therefore able to focus more on producing realistic and visually appealing images. The combined use of both image and text reward functions ($w.\ R_i\&R_t$) results in additional performance gains.

**Stable iteration process leads to further improvement.** The final enhancement comes from the incorporation of a momentum-based process reward function $R_m$, which takes into account historical improvements over time. This mechanism encourages the model to maintain a steady and consistent progression in its learning process. It draws parallels with human cognitive processes, where sustained, incremental progress is highly valued. By emphasizing gradual improvement, the momentum-based reward ensures that the model continuously refines its understanding and generation abilities over time, rather than oscillating between high-performance peaks and low-performance troughs. This method not only leads to enhanced model performance but also stabilizes the training trajectory, making the reinforcement learning process more reliable and robust in the long term.

## 4.4 Visualization of Multimodal-COT

The visualizations in Figure 4 demonstrate the adaptive nature of Vinci, which allows the model to autonomously control the length of the reasoning process. This capability enables the model to directly produce the final image after a single round of thought for simpler cases, while more complex scenarios may require two to three iterations of contemplation to yield the desired output. When tasked with creating an image of a blue vase and a black banana, the model's initial attempt missed the banana and had an unappealing vase shape. Recognizing these shortcomings, the model engaged in further reasoning to introduce the banana and enhance the vase's aesthetics. It continued this process until it achieved an image that not only met the user's specifications but also exhibited high visual quality.

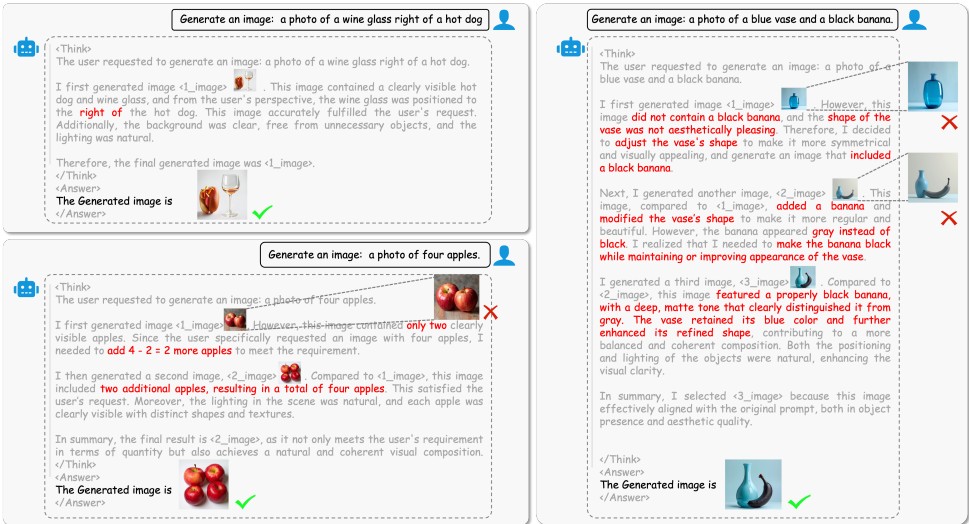

Figure 4: Case study demonstrating the variable-length MCoT generation process. For each query, the model iteratively generates and refines images based on feedback from the reasoning process.

This variable-length reasoning approach enhances both the output's fidelity and the process's overall efficiency. By allowing the model to determine the appropriate depth of thought required for each task, it ensures that while complex images receive the detailed consideration needed to meet high standards, simpler images are generated swiftly without unnecessary computational overhead.

# 5 Limitation and Future Work

There remains considerable room for exploration in the unified model for multimodal understanding and generation. Currently, the community lacks a truly general-purpose foundational model that can seamlessly integrate cross-modal understanding and generation capabilities. Although state-of-the-art models like Janus-Pro strive to balance these capabilities, they sacrifice a unified representation of understanding and generation at the foundational level. Additionally, the length of image tokens (which often requires hundreds or more tokens to represent a single image), combined with the limitations of the context window in autoregressive models, restricts the depth and efficiency of the model's reasoning capabilities. Future work can focus on improving both the understanding and generation capabilities of unified models, as well as developing better reasoning strategies to optimize efficiency and performance.

# 6 Conclusion

In this paper, we introduced Vinci, a novel image generation model that integrates deep reasoning capabilities through the use of interleaved multimodal Chain-of-Thought (MCoT). Unlike previous models that primarily focus on either image generation or understanding, Vinci achieves a unified approach that combines these two tasks in a way that allows the model to reflect on and iteratively improve its generated outputs in real time. Our method leverages reinforcement learning, where the model learns to refine its output based on the feedback from reward model. During the iterative generation process, we introduce a momentum-based process reward function that effectively balances exploration and exploitation. Experimental results demonstrate that Vinci significantly improves both image generation and instruction-following abilities, achieving a remarkable +22% performance improvement over the base model. This establishes Vinci as a new paradigm for reasoning-centric generative systems.

# 7 Acknowledgements

This work was supported by the "Pioneer" and "Leading Goose" R&D Program of Zhejiang (2025C02022) and the National Natural Science Foundation of China (No.62037001, No.62307032).

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

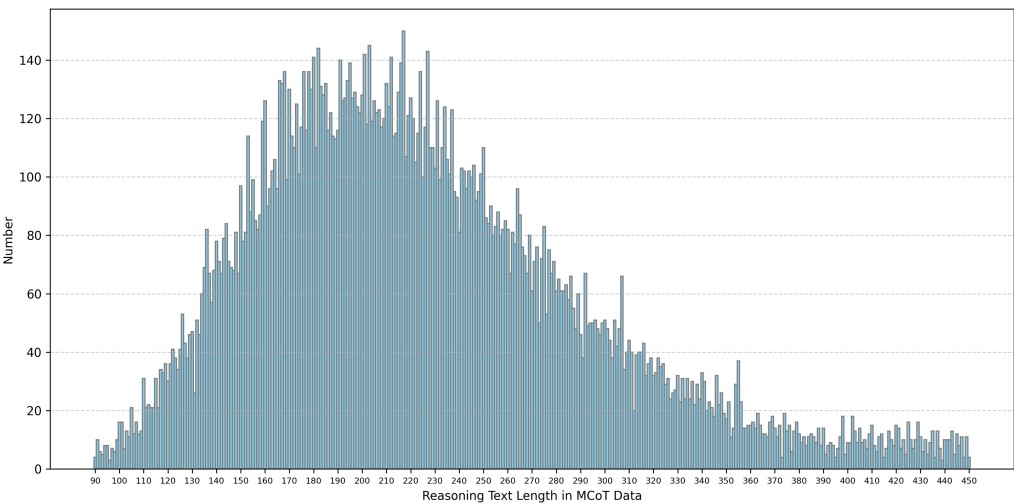

Figure 5: Distribution of reasoning text lengths in our MCoT dataset

# A    Dataset Construction

## A.1    MCoT Construction Pipeline

To construct our Multimodal-CoT(MCoT) dataset, we designed a multi-stage pipeline comprising Image Generation for MCoT, Text Generation for MCoT, and MCoT Construction. Given a text query $q$, we first generated a set of $n$ candidate images $\{p_1, p_2, ..., p_n\}$ using FLUX.1-dev [38]. To identify objects, we applied Mask2Former [7] to each image, yielding detection outputs $o_i$ and forming the set of tuples $\{(p_1, o_1), (p_2, o_2), ..., (p_n, o_n)\}$, where $o_i$ denoted the objects detected in image $p_i$. Then, to mitigate ambiguity in multi-image interpretation, we tasked Qwen-VL [68] with independently evaluating each tuple $(p_i, o_i)$ for its semantic alignment with the original query and overall quality. The detection result $o_i$ was explicitly provided to the model to reduce hallucinations. During this evaluation, the model produced a caption $c_i$ and a quality score $g_i$, resulting in annotated triples $(p_i, c_i, g_i)$. These were sorted in ascending order by $g_i$ to yield a ranked sequence of annotated samples. Finally, to construct the MCoT data, we randomly selected $k$ triples from the sorted list while preserving their relative score order. Each selected set must contain at least one high-quality (i.e., high-scoring) triple, which served as the final reasoning step. The resulting combination $\{(p_i^1, c_i^1, g_i^1), ..., (p_i^k, c_i^k, g_i^k)\}$ was input into Qwen-VL to construct the final MCoT data. The prompts used by Qwen-VL during the construction process are shown in Table 4.

The final MCoT dataset, consisting of 20,000 samples, comprises three types: 1-step MCoT, 2-step MCoT, and 3-step MCoT, which correspond to reasoning processes involving 1, 2, and 3 steps before generating the final image, respectively. Examples of these three types are shown in Table 3.

## A.2    Distribution of Reasoning Text Lengths

A key characteristic of MCoT data is the ability to clearly understand and articulate the content of images generated throughout the reasoning process. Therefore, the length of the accompanying text is an important aspect to consider. The distribution of reasoning text lengths in our constructed MCoT data is shown in Figure 5.

# B    QA-based Reward for Generated Text

In our research, we designed a QA-based reward function to evaluate the quality of the generated text in the context of text-to-image generation. This reward function assesses whether the generated text accurately describes the generated image, identifies any issues in the image, and provides strategies for improvement. In Table 5, we provide examples of the prompts used to guide the evaluation process.

| MCoT Data Type | Example |
|---|---|
| 1-step MCoT | `<think>`
The user asked to generate an image: a photo of two tennis rackets. One possible image was `<1_image>`. The image clearly reflected the prompt, showing two rackets placed side by side with distinct yet harmonious designs. The rackets were rendered in sharp detail, with red and black colors creating a vivid contrast. A soft, neutral background kept focus on the objects. Lighting and composition gave the image a polished, professional look, making it a faithful and visually appealing realization of the prompt. Overall, `<1_image>` was an excellent realization of the prompt, and I selected it as answer because it not only met the basic requirements but also elevated the concept through its high-quality execution and attention to detail.
`</think>`
`<answer><1_image></answer>` |
| 2-step MCoT | `<think>`
The user asked to generate an image: a photo of a sandwich below a sports ball. One possible image was `<1_image>`. While `<1_image>` did present a sandwich with a sports ball above it, the sandwich appeared somewhat simplistic and lacked detailed texturing. I needed to generate an image that clearly showed a sandwich below a sports ball, with the sandwich rendered in a more realistic and aesthetically pleasing way.
Then, I generated `<2_image>`, which offered a more refined and visually compelling representation of the prompt. The image clearly showed a sports ball above a sandwich, and their spatial relationship was accurate. The sandwich in `<2_image>` had a more realistic texture, with a visibly toasted bun that added depth and detail. The layers, including lettuce and other fillings, were vibrant and well-defined.
In summary, `<2_image>` correctly depicted a sandwich below a sports ball with a clear spatial arrangement. Both the sandwich and the sports ball featured improved textures and fine-grained details, contributing to the overall quality. These factors made `<2_image>` the preferred choice over `<1_image>`.
`</think>`
`<answer><2_image></answer>` |
| 3-step MCoT | `<think>`
The user asked to generate an image: a photo of an orange backpack and a green spoon. One possible image was `<1_image>`. However, it did not include an orange backpack as expected, which was a critical element of the prompt. I needed to generate an image that included an orange backpack.
I then generated `<2_image>`, which included the required orange backpack and aligned better with the prompt. While `<2_image>` was an improvement, its color scheme was relatively dull, and the background did not effectively complement the orange backpack. I needed to make the colors more vibrant.
Finally, I generated `<3_image>`, which featured a more vibrant and consistent color palette. The orange backpack and green spoon stood out clearly against a light green background. The lighting was evenly distributed, enhancing the textures and colors of the objects, which appeared more realistic and detailed.
In summary, I selected `<3_image>` as the final result because it accurately captured both the orange backpack and the green spoon with a clear and visually pleasing composition.
`</think>`
`<answer><3_image></answer>` |

Table 3: Introduction of the three types of MCoT data used in our cold-start stage. `<1_image>`, `<2_image>`, and `<3_image>` represent images generated after 1, 2, and 3 rounds of reasoning, respectively. The corresponding visual tokens are omitted for brevity.

In our implementation, we used a multi-modal large language model (MLLM) to evaluate the generated text based on the prompts provided above. The scores and explanations provided by the MLLM were then used as part of the overall reward function to guide the training of our text-to-image generation model. By incorporating this QA-based reward function, we aimed to enhance the model's ability to generate high-quality, contextually accurate, and self-reflective image descriptions, thereby improving the overall performance of the text-to-image generation process.

## C  Implementation Details

### C.1  Training Details

For our training, we adopted Emu3-Gen [71] as the base model, which has a unified image representation. The model's context length was extended to 15,360 tokens, which allowed for processing longer sequences of text and image pairs and supported approximately three iterations within each context window. The learning rate was set to 1e-5 and a beta of 0.01.

During the training, we utilized 16 A800 GPUs. The training was divided into two stages. In the first stage, known as the cold start, we set the batch size to 64 and trained for approximately 20 hours, allowing the model to learn the fundamental features and patterns of the data. In the second stage,

| Stage | Prompt Example |
|---|---|
| Text Generation | You are tasked with generating a caption for a given image and its detection results, and evaluating how well the image aligns with the original prompt in terms of semantic accuracy and generation quality.

Your task involves the following steps:
1. Caption Generation
- Generate a coherent and informative caption that accurately describes the given image, using the provided object detection results as a reference.
- The caption should cover key visual elements, including object types, positions, colors, and spatial relationships, and reflect the intent of the original prompt.

2. Prompt-Image Alignment and Generation Quality Evaluation
- Evaluate how well the image itself matches the original prompt in terms of semantic content.
- In addition, assess the visual quality of the generated image, including realism, clarity, and overall aesthetic quality.

3. Scoring
- Provide an overall score on a scale from 0 to 4 based on both prompt alignment and image quality:
- 0 indicates the image does not align with the prompt and is of low visual quality. It fails both semantically and aesthetically.
- 1 indicates the image is visually decent but fails to capture the core semantics of the prompt. It may contain hallucinated or unrelated content.
- 2 indicates the image is partially aligned with the prompt and has moderate quality. Some key elements may be missing or inaccurately rendered.
- 3 indicates the image correctly reflects the prompt but suffers from low visual quality (e.g., blurry, distorted, or unnatural rendering).
- 4 indicates the image is fully aligned with the prompt and of high visual quality. It accurately presents all required elements in a realistic, clear, and aesthetically pleasing manner. |
| MCoT Construction | You are given a sequence of (image, caption, score) triples ranked in ascending order by their score and their original prompt, where each triple consists of:
- an image generated based on a text prompt,
- a caption describing the image,
- a score indicating how well the image aligns with the original prompt in terms of semantic relevance and generation quality.

Your task is to simulate a step-by-step reasoning process that leads to the final decision about which image best satisfies the original prompt. This process should reflect how a human might evaluate and revise image generations based on feedback and visual inspection.

Please proceed as follows:
1. Analyze each image and caption in order, reflecting on what aspects are missing, incorrect, or can be improved.
2. Describe how the reasoning evolves across steps and why one image is better than the previous ones.
3. End the reasoning by selecting the best image and briefly summarizing why it is the final choice.

Output:
Your response must strictly follow the format below:

<think>
The user asked/requested to generate an image: [prompt]. One possible image is `<1_image>` [your generation and reasoning process]
</think>
<answer>[your choice]</answer> |

Table 4: Introduction of prompts used in MCoT construction

which involved reinforcement learning, we configured the group size to 8 and trained for 60 hours, further optimizing the model's performance through external feedback mechanisms.

## C.2 Data Preparation

All of our images are in the resolution of $512 \times 512$, and vision tokens were generated using vision tokenizer of Emu3. Following the Emu3 design, we incorporated five special tokens to merge textual and visual data, constructing interleaved vision-language MCoT data and document-like inputs for the training process.

Taking the 2-step MCoT data as an example, whenever `<1_image>` or `<2_image>` appears for the first time in the MCoT sequence, it is immediately followed by the corresponding image token block. The resulting training data can be structured as follows:

```
<1_image>[BOI]{meta text}[SOT]{vision tokens}[EOL][EOF][EOI]
```

| Example |
| --- |
| You are tasked with evaluating the quality of an image description generated by a text-to-image model. Please provide a detailed evaluation based on the following aspects:

1. Completeness of the Image Description:
- Assess whether the description covers all the key elements and details present in the image. Consider whether it includes descriptions of objects, their positions, colors, and any other relevant visual attributes.

2. Identification of Issues in the Image:
- Determine if the description identifies any discrepancies or issues in the generated image compared to the original prompt. This could include missing elements, incorrect colors, misplaced objects, or any other inconsistencies.

3. Strategies for Improvement:
- Suggest specific strategies or adjustments that could improve the accuracy and quality of the generated image. This could involve changes to the prompt or other recommendations to enhance the alignment between the generated image and the original prompt.

Image Description: [Generated Image Description]

Please provide a score for each aspect on a scale of 0 to 2, where:
- 0 indicates poor performance,
- 1 indicates average performance,
- 2 indicates excellent performance.
Additionally, provide a brief explanation for each score to justify your evaluation. |

Table 5: Introduction of the prompt used to evaluate the generated text's quality.

Here, the image token block begins with `[BOI]`, where {`meta text`} contains information about the image resolution. The token `[SOT]` marks the beginning of the vision token sequence. Additionally, `[EOL]` and `[EOF]` are inserted into the token stream to indicate line breaks. The image token block ends with `[EOI]`.

## D    Societal Impacts

Vinci, as a novel framework that integrates deep reasoning capabilities into text-to-image generation through a unified model with reinforcement learning, has the potential to significantly enhance various applications that rely on visual content creation [42, 43, 54, 66, 69, 80] and understanding [20, 21, 33, 34, 40, 44, 46, 47, 67, 73, 74, 76, 81]. One of the key positive impacts is the improvement in the quality and alignment of generated images with textual prompts. This capability can greatly benefit educational and creative industries. For instance, in education, Vinci can generate highly accurate and contextually relevant illustrations for textbooks [27, 45, 48, 70], making learning materials more engaging and accessible for students. In creative industries, such as graphic design and advertising, Vinci can assist designers in quickly generating high-quality visual concepts, thereby accelerating the creative process and potentially leading to more innovative and diverse visual content.

Despite its potential benefits, Vinci also poses several risks that need to be carefully considered. One of the primary concerns is the potential misuse of the technology for generating misleading or harmful visual content. Vinci's ability to generate high-quality images based on textual prompts increases the risk of creating deepfakes or manipulated images that could be used to spread disinformation, manipulate public opinion, or harm individuals' reputations. The ease with which these images can be generated and disseminated poses a significant threat to societal trust and information integrity.

