# OpenReview forum: "Vinci: Deep Thinking in Text-to-Image Generation using Unified Model with Reinforcement Learning"
_NeurIPS.cc/2025/Conference — NeurIPS 2025 poster_

### Official Review · Reviewer_CRiu · 2025-06-01

**Clarity:** 3
**Significance:** 3
**Originality:** 3
**Rating:** 4
**Confidence:** 4

**Summary:**

The paper introduces a method to enhance the text-to-image generation capabilities of multimodal models that utilize discrete tokens with causal attention for both image and text generation, such as Emu3-Gen. Specifically, the authors incorporate multimodal chain-of-thought (MCoT) reasoning and self-reflection capabilities, enabling the model to progressively refine image generation during inference. Initially, the model undergoes supervised fine-tuning on synthetically generated MCoT data, followed by reinforcement learning guided by three reward functions: a program-based reward leveraging detection models for image evaluation, a question-answering reward using multimodal large language models (MLLM) to assess generated textual reasoning, and a momentum process reward that dynamically encourages incremental improvements based on the improvements across images in the MCoT.

The authors demonstrate the effectiveness of their method by achieving a significant improvement of +22 absolute points (from 54% to 76%) on the GenEval benchmark. The key contributions include the focus of MCoT and self-reflection into text-to-image generation, developing a pipeline for synthesizing MCoT training data, designing reward functions for reinforcement learning in text-to-image, and providing ablation studies to quantify the impact of each reward component.

**Questions:**

Questions and Suggestions for the Authors:

1. Could you provide detailed performance metrics of the model immediately after the supervised fine-tuning (SFT) phase, prior to applying reinforcement learning (RL)? Including this information would significantly clarify the contribution of the RL phase.

2. How does your MCoT-based method compare to a simpler baseline strategy that involves selecting the best image from multiple generations based solely on the image-based reward models? The baseline should be tuned on high quality generations that were collected and used during the cold-start phase to level the playing field of collecting generations using flux and the scoring and filtering them. Including this information can clarify the benefits of using the MCoT framework.

3. Can you provide evidence or additional experiments demonstrating the model's ability to generalize its reasoning capabilities to novel tasks or attributes not explicitly covered during training? For example, if specific reasoning tasks (such as color attribution) were excluded from training, does the model effectively adapt and correctly handle these tasks during inference? This again, can highlight the benefits of using the MCoT framework.

4. Could you expand upon the implementation details provided in the paper? Clarifying specific technical choices (e.g., hyperparameter selection, computational resources, runtime considerations) would improve reproducibility.

**Ethical Concerns:**

["NO or VERY MINOR ethics concerns only"]

**Final Justification:**

The rebuttal response answered my concerns and therefore I increase the rating to borderline accept.

**Limitations:**

The limitations section could be expanded further by explicitly discussing specific scenarios or failure cases where the proposed method and model might underperform or fail entirely.

**Paper Formatting Concerns:**

ok

**Quality:**

2

**Strengths And Weaknesses:**

Strengths:

1. The primary research question, which explores the use of multimodal chain-of-thought (MCoT), is compelling and worthwhile to investigate.

2. The pipeline developed to synthesize MCoT data for supervised fine-tuning (SFT) can be beneficial for future research.

3. The insights regarding the contribution of the text-based reward model to image generation performance are valuable and interesting, and highlight the potential of multimodal models.

4. The ablation study, which analyzes the impact of different reward model combinations, is informative.

Weaknesses:

1. The authors do not clearly present results immediately following the cold-start phase. Given the substantial high-quality SFT dataset collected for the specific domain, one would expect a significant performance improvement after SFT alone. Therefore, a more appropriate baseline to clearly demonstrate the impact of the reward models would be the model’s performance after the SFT phase.

2. Despite being central to the paper's focus, the efficacy of MCoT as a method remains somewhat uncertain, particularly considering its computational demands. A simpler baseline comparison could involve selecting the best image from multiple generations using image-based reward models.  Additionally, a valuable comparison would involve performing SFT on top-scoring images and then selecting the best result from several generations, to directly assess whether MCoT offers significant advantages over simpler alternatives.

3. The generalization capability of the model is unclear. A significant advantage of chain-of-thought methods in large language models is their ability to learn reasoning processes that generalize to novel tasks and scenarios. It remains uncertain whether this model can generalize beyond its specific training data. For example, if the SFT data and prompts during RL would not include prompts with color attribution, it is unclear if the model will learn to adjust color attributes effectively during inference.

4. The implementation details provided in the paper are somewhat limited and could benefit from further elaboration to facilitate clearer understanding and reproducibility.

---

> ### Author Rebuttal · Authors · 2025-07-31
>
> Thank you for your valuable feedback. We address each of your points individually below.
>
>
> ### **Q1. The results of SFT's model**
>
> Thank you for the insightful suggestion. We have accordingly supplemented Supervised Fine-Tuning (SFT) results, presented in the table below. While SFT yields a modest improvement **(+7)**, the dominant gain **(+15)** emerges from the reinforcement-learning phase. This outcome substantiates that RL is **significantly more effective** than SFT in transferring the model’s visual-understanding capabilities into the image-generation process.
>
> |                | **single obj** | **Two obj** | **Counting** | **Colors** | **Position** | **Color Attr.** | **Overall** |
> | :------------: | :------------: | :---------: | :----------: | :--------: | :----------: | :-------------: | :---------: |
> |    Emu3-Gen    |      0.98      |    0.71     |     0.34     |    0.81    |     0.17     |      0.21       |    0.54     |
> | Emu3-Gen(SFT ) |      0.98      |    0.74     |     0.39     |    0.81    |     0.50     |      0.24       |    0.61     |
> |     Vinci      |      0.99      |    0.86     |     0.48     |    0.83    |     0.86     |      0.54       |    0.76     |
>
> ### **Q2. Compare with Best of N**
>
> We have additionally compared Vinci against Best-of-N baselines; Where Emu3(w.sf&Best-of-N) is fine-tuned with the same cold start data and generated with the Best-of-N (N=16) strategy. The results, summarized in the table below, show that Vinci retains **a clear advantage**. Best-of-N merely re-ranks fixed outputs without enhancing the generator itself, whereas our RL formulation transfers the model’s intrinsic understanding directly into the generation process, yielding **sustained improvements** in quality.
>
> |                         | **single obj** | **Two  obj** | **Counting** | **Colors** | **Position** | **Color Attr.** | **Overall** |
> | :---------------------: | :------------: | :-------: | :----------: | :--------: | :----------: | :-------------: | :---------: |
> |         SDXL[1]         |       -        |     -     |      -       |     -      |      -       |        -        |    0.65     |
> |     Emu3-Gen(SFT )      |      0.98      |   0.74    |     0.39     |    0.81    |     0.50     |      0.24       |    0.61     |
> | Emu3(w.SFT & Best-of-N) |      0.99      |   0.79    |     0.43     |    0.84    |     0.59     |      0.34       |    0.66     |
> |          Vinci          |      0.99      |   0.86    |     0.48     |    0.83    |     0.86     |      0.54       |    0.76     |
>
> >[1]A General Framework for Inference-time Scaling and Steering of Diffusion Models
>
> ### **Q3. The generalization capability**
>
> Thank you for the insightful suggestion; we have conducted the requested experiment, with results presented in the table below. The model achieves **50 %** accuracy on color_attr which never explicitly seen during training, and the performance is close to Vinvi **(54 %)**, demonstrating that the MCoT framework **successfully generalizes** its reasoning capabilities to novel visual properties.
>
> |                              | **single obj** | **Two  obj** | **Counting** | **Colors** | **Position** | **Color Attr.** | **Overall** |
> | :--------------------------: | :------------: | :-------: | :----------: | :--------: | :----------: | :-------------: | :---------: |
> |             Emu3             |      0.98      |   0.71    |     0.34     |    0.81    |     0.17     |      0.21       |    0.54     |
> |  Emu3(SFT) (w.color_attr）   |      0.98      |   0.76    |     0.43     |    0.81    |     0.53     |      0.28       |    0.63     |
> | Emu3(SFT) (w/o.color_attr）  |      0.98      |   0.75    |     0.43     |    0.79    |     0.53     |      0.22       |    0.62     |
> | Emu3(SFT+RL)(w/o.color_attr) |      0.99      |   0.83    |     0.49     |    0.80    |     0.85     |    **0.50**     |    0.74     |
> |            Vinci             |      0.99      |   0.86    |     0.48     |    0.83    |     0.86     |    **0.54**     |    0.76     |
>
> ### **Q4. More implementation details**
>
> To accommodate longer interleaved sequences, we extend the model's context length to 15,360 tokens. Each image is 512×512 in resolution, corresponding to 4,096 visual tokens.
>
> The training is conducted on 16 A800 GPUs, and consists of two stages:
>
> ● In the cold start stage, we use a batch size of 64, a learning rate of 1e-5, and weight decay β=0.01. This stage lasts approximately 20 hours, allowing the model to acquire basic multimodal generation capabilities.
>
> ● In the reinforcement learning stage, we set the group size to 8 and train for an additional 60 hours, refining model performance through reward-based optimization.
>
> Please refer to the supplementary material for more details, and we will also publish the code and data to ensure reproducibility of the work.

---

> > ### Comment · Reviewer_CRiu · 2025-08-03
> >
> > I thank the authors for the rebuttal response and raised my score.

---

> > > ### Author Response · Authors · 2025-08-03
> > >
> > > Thank you very much for your positive feedback and for raising your score—we truly appreciate your support and the time you devoted to reviewing our work.

---

### Official Review · Reviewer_2Qyn · 2025-06-30

**Clarity:** 2
**Significance:** 2
**Originality:** 2
**Rating:** 3
**Confidence:** 3

**Summary:**

The paper introduces Vinci, a unified autoregressive text-to-image model that interleaves image generation and understanding through a Multimodal Chain-of-Thought (MCoT). After a small “cold-start” finetuning with 20 k synthetic MCoT traces, the authors apply reinforcement learning guided by three rewards—program-based visual checks, QA-based textual assessment, and a novel momentum reward—to let the model iteratively critique and refine its own outputs. On the object-focused GenEval benchmark, Vinci improves the Emu3-Gen base by +22 % overall, narrowing the gap with much larger systems while remaining fully end-to-end.

**Questions:**

What is the training cost and inference cost of the model compared with direct or simple cot?

**Ethical Concerns:**

["NO or VERY MINOR ethics concerns only"]

**Limitations:**

See weaknesses

**Quality:**

2

**Strengths And Weaknesses:**

Strengths

1. Conceptual novelty – The work realizes interleaved MCoT inside a single autoregressive generator, enabling the model to “look at” intermediate images and revise them on-the-fly. This directly tackles the reflection bottleneck of previous unified methods.

2. Reward design – The momentum-based reward elegantly couples short-term quality with long-term improvement stability, an idea reminiscent of adaptive optimisers yet new to RLHF for vision–language models.

3. Thorough empirical evidence – Results span benchmark scores, qualitative rollouts and an ablation that disentangles each reward’s contribution, convincingly supporting the claimed +22 % gain and showing Vinci to be competitive with much larger models.

Weaknesses
1. Limited evaluation scope – All quantitative claims rest on GenEval; there is no open-domain or human preference study, leaving generalisation and perceptual quality uncertain.

2. Heavy reliance on automatically-synthesised MCoT traces. The “cold-start” stage uses just 20 k traces whose images come from Flux and captions from Qwen-VL. Because neither source is curated or human-checked, the traces may carry style biases or captioning errors that the policy then imitates, limiting robustness outside the synthetic data distribution.

3. Both the program-based visual check and the QA-based textual check depend on external detectors or MLLMs. Any failure—e.g., detector mis-counting objects in cluttered scenes—directly contaminates the reinforcement signal, encouraging reward hacking rather than genuine image-text alignment.

4. While the paper defines α, λ, γ for the momentum term, it never reports the chosen values or provides a sensitivity study.

---

> ### Author Rebuttal · Authors · 2025-07-31
>
> Thank you for your valuable feedback. We address each of your points individually below.
>
>
>
> **Q1. Expand the evaluation scope**
>
> We thank the reviewers for this suggestion. Owing to the tight rebuttal schedule, we engaged three undergraduate annotators to conduct a human evaluation on **Geneval**; the results are summarized in the table below. Despite the inevitable noise in automatic metrics, the human judgements confirm that **Vinci** still outperforms all baselines.
>
> Additionally, we have now reported results on **DPG-Bench** and **T2I-CompBench**: Vinci ranks second on DPG-Bench—only marginally behind Janus-Pro-7B—and achieves the highest scores on T2I-CompBench, manifesting a substantial gain over Emu3. Taken together, these supplementary benchmarks further corroborate Vinci’s effectiveness.
>
> **Table 1. Human Evaluation**
>
> | Model      | Single Obj | Two Obj | Counting | Colors | Position | Color Attr. | Overall |
> |------------|------------|---------|----------|--------|----------|--------------|---------|
> |*Model-Based Evaluation*
> | Show-o     | 0.98       | 0.80    | 0.66     | 0.84   | 0.31     | 0.50         | 0.68    |
> | Emu3-Gen   | 0.98       | 0.71    | 0.34     | 0.81   | 0.17     | 0.21         | 0.54    |
> | T2I-R1     | 0.99       | 0.77    | 0.70     | 0.87   | 0.41     | 0.30         | 0.67    |
> | Vinci      | 0.99       | 0.86    | 0.48     | 0.83   | 0.86     | 0.54         | 0.76    |
> |*Human Evaluation*
> | Show-o     | 0.97       | 0.81    | 0.64     | 0.83   | 0.29     | 0.49         | 0.67    |
> | Emu3-Gen   | 0.98       | 0.69    | 0.35     | 0.81   | 0.18     | 0.18         | 0.53    |
> | T2I-R1     | 0.98       | 0.75    | 0.72     | 0.86   | 0.37     | 0.28         | 0.66    |
> | Vinci      | 0.99       | 0.82    | 0.47     | 0.78   | 0.88     | 0.48         | 0.74    |
>
>
>
> **Table 2.  DPG-Bench**
>
> | Model         | Overall | Global | Entity | Attribute | Relation | Other  |
> |---------------|---------|--------|--------|-----------|----------|--------|
> | **Gen. Only** |
> | SD3-Medium    | 84.08   | 87.90  | 91.01  | 88.83     | 80.70    | 88.68  |
> | Flux.1 dev    | 84.00   | 82.10  | 89.50  | 88.70     | 91.10    | 89.40  |
> | **Unified.**  |
> | Show-o        | 67.27   | 79.33  | 75.44  | 78.02     | 84.45    | 60.80  |
> | Janus-Pro-7B  | 84.19   | 86.90  | 88.90  | 89.40     | 89.32    | 89.48  |
> | Emu3          | 80.60   | 85.21  | 86.68  | 86.84     | 90.22    | 83.15  |
> | Vinci-sft     | 81.18   | 86.12  | 86.91  | 87.53     | 90.64    | 82.98  |
> | Vinci         | 82.54   | 87.68  | 88.03  | 87.72     | 91.02    | 85.43  |
>
> **Table 3. T2I-CompBench**
>
> | Model         | Color   | Shape  | Texture | Spatial | Non-Spatial | Complex |
> |---------------|---------|--------|---------|---------|-------------|---------|
> | **Gen. Only** |
> | SD3-Medium    | 0.8132  | 0.5885 | 0.7334  | 0.3200  | 0.4084      | 0.3771  |
> | Flux.1 dev    | 0.7407  | 0.5718 | 0.6922  | 0.2863  | 0.3127      | 0.3703  |
> | **Unified.**  |
> | Show-o        | 0.5600  | 0.4100 | 0.4600  | 0.2000  | 0.3000      | 0.2900  |
> | Janus-Pro-7B  | 0.6359  | 0.3528 | 0.4936  | 0.2061  | 0.3085      | 0.3559  |
> | T2I-R1        | 0.8130  | 0.5852 | 0.7243  | 0.3378  | 0.3090      | 0.3993  |
> | Vinci-sft     | 0.6274  | 0.4729 | 0.6189  | 0.2292  | 0.2711      | 0.3752  |
> | Vinci         | 0.8225  | 0.6062 | 0.7374  | 0.3243  | 0.3148      | 0.4081  |
>
> ---
>
> **Q2. About the MCoT data**
>
> We decompose data collection into sequentially verifiable sub-tasks. So we can interleave human inspection at each stage, yielding lower cumulative error than any single end-to-end model that lacks such intervention points.
>
> Although we did not conduct a thorough manual inspection, we used GPT-4 to evaluate and filter out data that contains hallucinations or failures at the MCoT endpoints.
> Empirically, sampling indicates that both Qwen-VL and GPT-4o exhibit strong and robust comprehension of the generated images, and we have not observed any systematic failures or notable shortcomings.
>
> Due to space constraints we cannot include external links in this rebuttal, but the full dataset and codebase will be released concurrently with the camera-ready version to enable reproducibility and future extensions.
>
> ---
>
> **Q3. About the reward model**
>
> We acknowledge that detector failures can corrupt the reinforcement signal; however, the human evaluation reported in Q1 shows that such instances are not too much.
>
> Developing more robust reward models is undeniably important, yet it lies beyond the scope of this paper and is therefore deferred to future research.
>
> ---
>
> **Q4. The hyperparameters for momentum reward**
>
> In our actual implementation, we have $\alpha = 0.5$, $\lambda = 0.5$, and $\gamma = 0.5$.
> Here:
> - $\alpha$ balances the single image quality score and momentum update;
> - $\lambda$ is the time decay factor, encouraging improvement at earlier steps;
> - $\gamma$ controls the sensitivity of $V_k$ to recent changes (smaller = more sensitive).
>
> Due to time constraints, we additionally conducted two sets of sensitivity studies, shown below:
>
> **Table 4. Hyperparameters Sensitivity Studies**
>
> | Setting                 | Single Obj | Two Obj | Counting | Colors | Position | Color Attr. | Overall |
> |------------------------|------------|---------|----------|--------|----------|--------------|---------|
> | $\alpha=\lambda=\gamma=0.1$ | 0.99       | 0.83    | 0.51     | 0.80   | 0.84     | 0.55         | 0.75    |
> | $\alpha=\lambda=\gamma=0.8$ | 0.99       | 0.85    | 0.50     | 0.82   | 0.85     | 0.55         | 0.76    |
> | $\alpha=\lambda=\gamma=0.5$ (Ours) | 0.99       | 0.86    | 0.48     | 0.83   | 0.86     | 0.54         | 0.76    |
>
> The results indicate that these hyperparameters have relatively little impact on the outcome.
>
> ---
>
> **Q5. The training cost and inference cost**
>
> - **Cold-start stage**: batch size = 64, learning rate = 1e-5, weight decay $\beta = 0.01$
>   Duration: ~20 hours, to acquire basic multimodal generation capability.
>
> - **Reinforcement learning stage**: group size = 8, training time = 60 hours
>   Purpose: refine performance via reward-guided optimization.
>
> **Table 5. Inference Cost**
> | Model      | Inference Type | Image Samples | Time (min) |    Memory (GB)  | Mem. (%) | Time (%) |
> |------------|----------------|----------------|-------------|------------|----------|----------|
> | Emu3-Gen   | Direct T2I     | 1              | 5.00        | 23         | 100.00   | 100.00   |
> | Vinci      | MCoT           | 1              | 5.87        | 24         | 104.35   | 117.33   |
> | Vinci      | MCoT           | 2              | 6.27        | 27         | 117.39   | 125.33   |
> | Vinci      | MCoT           | 3              | 6.80        | 28         | 121.74   | 136.00   |

---

### Official Review · Reviewer_5wxe · 2025-07-02

**Clarity:** 3
**Significance:** 3
**Originality:** 3
**Rating:** 5
**Confidence:** 4

**Summary:**

The authors introduce Vinci, a novel framework that integrates multimodal Chain-of-Thought (MCoT) reasoning with reinforcement learning (RL) to enhance text-to-image generation. Unlike existing unified models that treat image generation and understanding as separate tasks, Vinci enables iterative refinement of generated images through real-time self-reflection.

**Questions:**

1) Why exclude comparisons to RL-based T2I models like DALL-E 3? Including these would strengthen claims of superiority.

2) Provide specifics on the object detection model and GPT-4o filtering thresholds for MCoT data construction.

3) Derive or cite theoretical guarantees for Equation 5’s stability.

4) Quantify the computational overhead of iterative refinement, e.g., FLOPs per iteration.

**Ethical Concerns:**

["NO or VERY MINOR ethics concerns only"]

**Final Justification:**

The authors addressed all my concerns from Weaknesses and Questions, so I lifted my final score.

**Limitations:**

Yes

**Quality:**

3

**Strengths And Weaknesses:**

Strengths
1) The integration of MCoT with RL for iterative image refinement is novel. The momentum-based reward function is a creative solution to stabilize multi-round generation.
2) Comprehensive experiments on GenEval show significant gains (+22% over baselines), with detailed ablation studies validating each component’s contribution.
​​3)  The framework addresses a critical gap in unified models by enabling real-time self-correction, as shown in qualitative results.

Weaknesses
1) While the authors compare with unified models (e.g., Janus-Pro, GPT-4o), it omits comparisons to recent RL-based T2I models like DALL-E 3 or Stable Diffusion 3, which are industry standards.
​​2) The MCoT data pipeline lacks details on object detection models and GPT-4o filtering criteria. Hyperparameters for RL training are not justified.
3) The momentum reward’s theoretical guarantees (e.g., convergence) are not discussed, despite its central role.

---

> ### Author Rebuttal · Authors · 2025-07-31
>
> Thank you for your valuable feedback. We address each of your points individually below.
>
> **Q1. Compare with RL-based T2I models**
>
> We have added comparisons with recent RL-based text-to-image models (Tables 2 and 3 in [1]); the results are presented below. **Vinci** outperforms both RL-fine-tuned Stable Diffusion and FLUX, primarily because our pipeline incorporates a *self-reflective mechanism* during reinforcement learning that leverages the model’s own visual-understanding capabilities to enhance generation quality.
>
> | Model                                              | Geneval Score |
> |----------------------------------------------------|----------------|
> | Stable Diffusion with HPS-v2.1 & CLIP Score        | 0.52           |
> | FLUX with HPS-v2.1 & CLIP Score                    | 0.69           |
> | **Vinci**                                          | **0.76**       |
>
> ---
>
> **Q2. More details about MCoT data construction**
>
> For the object detection model, we use Mask2Former with a Swin-S backbone, following the official MMDetection configuration and pretrained on the COCO dataset. We apply a detection confidence threshold of 0.6 to remove low-confidence predictions. For computational efficiency and consistency, we retain at most 16 objects per class per image.
>
> During data filtering, we introduce GPT-4o as a final quality filter to identify hallucinations or incoherent reasoning chains. Specifically, for each constructed MCoT sample, we query GPT-4o with a structured prompt to evaluate:
>
> 1. Whether the final image is semantically aligned with the original prompt;
> 2. Whether the reasoning steps are logically coherent and progressively improve the image’s consistency or visual quality.
>
> GPT-4o provides a binary judgment ("pass"/"fail") along with a short explanation. We retain only the samples that pass this check and discard the rest. In practice, this filtering process removes approximately 8% of low-quality MCoT samples.
>
> ---
>
>
> **Q3. Derive theoretical guarantees for Equation 5’s stability**
>
> Below, we provide a comprehensive analysis of the reward function’s theoretical underpinnings.
>
>
> ### 1. Convergence Analysis
>
> **Finite Sequence Length ($n$):**
> For a finite sequence length $n$, all terms in the process reward function $R_m$ are finite sums, which guarantees that the value of $R_m$ remains finite and computable:
> $$
> R_m \text{ is bounded and computable for finite } n.
> $$
>
> **As $n \to \infty$:**
>
> a. The first term, which averages the quality scores $R_i(s_t)$, converges to the expected quality score, assuming a stable sequence:
> $$
> \lim_{n \to \infty} \frac{1}{n} \sum_{t=1}^{n} R_i(s_t) = \mathbb{E}[R_i(s_t)].
> $$
>
> b. The second term, which accounts for momentum-based improvement, converges depending on the decay factor $\lambda$:
>
> \begin{cases}
> \lambda < 1: & \sum_{k=1}^{\infty} \frac{\Delta_k \lambda^{k-1}}{\sqrt{V_k} + \epsilon} \leq \frac{1}{\epsilon(1 - \lambda)} \quad \text{(absolutely convergent)}
> \end{cases}
>
> \begin{cases}
> \lambda = 1: & \sum_{k=1}^{n-1} \frac{\Delta_k}{\sqrt{V_k} + \epsilon} \leq \frac{n - 1}{\epsilon} \quad \text{(bounded for finite $n$)}
> \end{cases}
>
>
> Thus, the reward function $R_m$ guarantees convergence as $n \to \infty$, with the sum of improvements tending to a stable value based on the decay rate $\lambda$.
>
> ---
>
> ### 2. Edge Case Testing
>
> The behavior of the reward function in edge cases demonstrates its adaptive nature and stability:
>
> - **No Improvement** ($\forall k, \Delta_k = 0$):
>   $$
>   R_m = \frac{1}{n} \sum_{t=1}^{n} R_i(s_t) \quad \text{(degenerates to the average quality score)}
>   $$
>
> - **Initial Large Improvement** ($R_i(s_0) = 0$, $R_i(s_1) = 1$):
>   $$
>   V_1 = 0.1, \quad \frac{\Delta_1}{\sqrt{V_1} + \epsilon} \approx 3.162 \quad \text{(modest amplification of early improvement)}
>   $$
>
> - **Low Variability Small Improvements** ($\Delta_k = 0.01$, $V_k = 10^{-6}$):
>   $$
>   \frac{\Delta_k}{\sqrt{V_k} + \epsilon} \approx 10 \quad \text{(adaptive amplification of consistent small gains)}
>   $$
>
> - **Constant Scores** ($\forall t,\ R_i(s_t) = c$):
>
>   $$
>   \Delta_k = 0 \Rightarrow R_m = c \quad \text{(no momentum contribution)}
>   $$
>
> When the quality scores are constant, the reward function focuses entirely on the absolute quality score, with no momentumcomponent.
>
> ---
>
> ### 3. Mathematical Stability Conclusions
>
> We formally summarize the stability guarantees of the momentum reward function $R_m$:
>
> - **Definition Completeness:**
>   $$
>   \sqrt{V_k} + \epsilon > \epsilon > 0 \quad \text{ensures no division by zero}.
>   $$
>
> - **Global Boundedness:**
>   $$
>   R_m \in \left[0,\ 1 + \alpha \cdot \min\left( \frac{1}{\epsilon(1 - \lambda)},\ \frac{n - 1}{\epsilon} \right) \right],
>   $$
>   ensuring $R_m$ remains within a finite and interpretable range.
>
> - **Numerical Robustness:**
>   Setting $\epsilon = 10^{-8}$ avoids underflow and overflow. All inputs are bounded:
>   $$
>   R_i(s_t) \in [0, 1], \quad \Delta_k \in [0, 1], \quad V_k \geq 0.
>   $$
>
> - **Convergence Guarantee:**
>   For $\lambda < 1$, the infinite sum of improvements is absolutely convergent:
>   $$
>   \lambda < 1 \Rightarrow \text{long-term stability and consistent reward computation}.
>   $$
>
> In conclusion, the momentum process reward function $R_m$ exhibits robust theoretical properties that ensure its stability and convergence during training.
>
>
> ---
>
> **Q4. The FLOPs per iteration**
>
> We measured the computational cost across different iterations; the results are shown in the table below. The overhead scales almost linearly with the number of generated images.
>
> | Model      | Inference Type | Image Samples | TFLOPs | TFLOPs (%) |
> |------------|----------------|----------------|--------|------------|
> | Emu3-Gen   | Direct T2I     | 1              | 25.3   | 100.00     |
> | Vinci      | MCoT           | 1              | 28.5   | 112.65     |
> |        Vinci    |          MCoT       | 2              | 56.9   | 224.11     |
> |     Vinci       |     MCoT            | 3              | 85.4   | 337.55     |

---

> ### Comment · Reviewer_5wxe · 2025-08-05
>
> Thank you very much for your response. I have read your response to my comments and other reviewers. The authors address my concerns.

---

> > ### Author Response · Authors · 2025-08-08
> >
> > Thank you for your thoughtful review of our rebuttal. We appreciate your constructive feedback

---

### Official Review · Reviewer_QUHK · 2025-07-03

**Clarity:** 2
**Significance:** 3
**Originality:** 2
**Rating:** 4
**Confidence:** 3

**Summary:**

This paper proposes to apply Multimodal Chain-of-Thought (MCoT) reasoning and Reinforcement Learning (RL) to text-to-image generation within a unified multimodal model, specifically Emu3. It follows the recent RL techniques in large-language models (LLMs), especially the Deepseek-R1. Specifically, it first collects a small amount of multimodal chain-of-thought data from GPT-4o, used for cold-start of the RL training. Then it introduces a momentum-based reward function that distributes process rewards based on historical improvements, which improves the long-term stability of RL optimization. This method is based on Emu3 because of its unified image representation. Experiments on GenEval have demonstrated +22% improvement compared to the base model.

**Questions:**

In addition to the Weaknesses, I have additional questions about the paper details.

1. This paper does not compare with other CoT & RL image generation models such as T2I-R1. While they may be different in the overall architecture, they address a similar problem. A direct comparison, possibly by applying T2I-R1 to Emu3 under similar settings, would strengthen the results.

2. Are there any methods that can be used to analyze or visualize the effectiveness of the momentum-based reward function and show that it stabilizes the training process?

**Ethical Concerns:**

["NO or VERY MINOR ethics concerns only"]

**Final Justification:**

Most of my concerns are addressed in the rebuttal. I raised the score towards accept.

**Limitations:**

yes

**Quality:**

3

**Strengths And Weaknesses:**

Strengths

1. This method explores applying Multimodal Chain-of-Thought (MCoT) and RL to cross-modal multimodal models, enabling iterative refinement over both image and text modalities. This is a step beyond prior unimodal CoT approaches, which focus only on textual prompt refinement.

2. It proposes a momentum-based reward function, which is a meaningful contribution that addresses the common instability issues in RL finetuning.

3. The results on GenEval have shown good improvement compared to Emu3.

Weaknesses

1. The method design depends on a unified autoregressive model that has unified text and image representations, which limits its application to different MLLMs.

2. The MCoT data is only collected by GPT-4o without real-world data, which will possibly be limited by GPT-4o's capabilities.

3. The evaluation is limited to GenEval, which contains short prompts and may not be enough to evaluate the CoT capabilities. Experiments on other benchmarks such as DPG-Bench from ELLA and T2I-CompBench should be included.

Reference: ELLA: Equip Diffusion Models with LLM for Enhanced Semantic Alignment

4. The ablation study only evaluates the effectiveness of reward functions. However, the alterations with/without the MCoT data should also be included to validate the effectiveness of the cold start phase.

---

> ### Author Rebuttal · Authors · 2025-07-31
>
> Thank you for your valuable feedback. We address each of your points individually below.
>
>
> **Q1. The limitations of the application**
>
> We contend that this should not be regarded as a limitation of our work. Recently, the community has devoted considerable attention to unified autoregressive frameworks [1, 2, 3] that represent text and images within a single modality; consequently, a growing family of such unified models has emerged. We align with the position articulated in [1] that *a truly unified autoregressive model should adopt a unified representation for both text and images*—without which it cannot claim to be a foundational unified system. Through this paper, we also hope to further encourage the community to reflect on and refine the architectural design of unified autoregressive models.
>
> > [1] Selftok: Discrete Visual Tokens of Autoregression, by Diffusion, and for Reasoning
>
> > [2] Chameleon: Mixed-Modal Early-Fusion Foundation Models
>
> > [3] X-Omni: Reinforcement Learning Makes Discrete Autoregressive Image Generative Models Great Again
>
>
> ---
>
> **Q2. Limited by GPT-4o's capabilities**
>
> We did not use GPT-4o to produce the MCoT annotations. High-quality human labels are prohibitively expensive to collect at scale; therefore, the current MCoT dataset pairs images generated by Flux with captions provided by Qwen-VL. GPT-4o is used to evaluate and filter out data that contains hallucinations or failures at the MCoT endpoints.
> Empirically, sampling indicates that both Qwen-VL and GPT-4o exhibit strong and robust comprehension of the generated images, and we have not observed any systematic failures or notable shortcomings. The complete dataset will be released concurrently—please stay tuned.
>
> ---
>
> **Q3. The results based on other benchmarks**
>
> Following your suggestion, we have added results on **DPG-Bench** and **T2I-CompBench**. Vinci ranks second on DPG-Bench—only marginally behind Janus-Pro-7B—and achieves the highest scores on T2I-CompBench, demonstrating a substantial improvement. These additional benchmarks further corroborate the effectiveness of Vinci.
>
> **Table 1. DPG-Bench**
>
> | Method        | Overall | Global | Entity | Attribute | Relation | Other  |
> |---------------|---------|--------|--------|-----------|----------|--------|
> | **Gen. Only** |
> | SD3-Medium    | 84.08   | 87.90  | 91.01  | 88.83     | 80.70    | 88.68  |
> | Flux.1 dev    | 84.00   | 82.10  | 89.50  | 88.70     | 91.10    | 89.40  |
> | **Unified.**  |
> | Show-o        | 67.27   | 79.33  | 75.44  | 78.02     | 84.45    | 60.80  |
> | Janus-Pro-7B  | 84.19   | 86.90  | 88.90  | 89.40     | 89.32    | 89.48  |
> | Emu3          | 80.60   | 85.21  | 86.68  | 86.84     | 90.22    | 83.15  |
> | Vinci-sft     | 81.18   | 86.12  | 86.91  | 87.53     | 90.64    | 82.98  |
> | Vinci         | 82.54   | 87.68  | 88.03  | 87.72     | 91.02    | 85.43  |
>
> **Table 2. T2I-CompBench**
>
> | Model         | Color   | Shape  | Texture | Spatial | Non-Spatial | Complex |
> |---------------|---------|--------|---------|---------|-------------|---------|
> | **Gen. Only** |
> | SD3-Medium    | 0.8132  | 0.5885 | 0.7334  | 0.3200  | 0.4084      | 0.3771  |
> | Flux.1 dev    | 0.7407  | 0.5718 | 0.6922  | 0.2863  | 0.3127      | 0.3703  |
> | **Unified.**  |
> | Show-o        | 0.5600  | 0.4100 | 0.4600  | 0.2000  | 0.3000      | 0.2900  |
> | Janus-Pro-7B  | 0.6359  | 0.3528 | 0.4936  | 0.2061  | 0.3085      | 0.3559  |
> | T2I-R1        | 0.8130  | 0.5852 | 0.7243  | 0.3378  | 0.3090      | 0.3993  |
> | Vinci-sft     | 0.6274  | 0.4729 | 0.6189  | 0.2292  | 0.2711      | 0.3752  |
> | Vinci         | 0.8225  | 0.6062 | 0.7374  | 0.3243  | 0.3148      | 0.4081  |
>
>
> ---
>
> **Q4. Ablation on MCoT**
>
> Thank you for the suggestion. We have conducted an ablation on the MCoT data:
>
> - **w. MCoT**: SFT with MCoT followed by RL.
> - **w/o MCoT & w. RL**: Skip SFT and train directly with RL, using only a format reward to learn the interleaved output pattern.
>
> Results show that the primary benefit of MCoT during cold-start is to support the interleaved output format; it yields only a modest improvement in generation quality (+7) compared with the larger gains delivered by RL (+15).
> For Emu3—whose native architecture lacks interleaved text–image capability—learning the format solely through the format reward is difficult, leading to a significant drop in final performance (overall score: 64).
>
>  **Table 3. MCoT Ablation Results**
>
> | Model                  | Single Obj | Two Obj | Counting | Colors | Position | Color Attr. | Overall |
> |------------------------|------------|---------|----------|--------|----------|--------------|---------|
> | Emu3-Gen               | 0.98       | 0.71    | 0.34     | 0.81   | 0.17     | 0.21         | 0.54    |
> | w. MCoT                | 0.98       | 0.74    | 0.39     | 0.81   | 0.50     | 0.24         | 0.61    |
> | w/o MCoT & w. RL       | 0.99       | 0.74    | 0.42     | 0.82   | 0.57     | 0.29         | 0.64    |
> | w. MCoT & RL           | 0.99       | 0.86    | 0.48     | 0.83   | 0.86     | 0.54         | 0.76    |
>
> ---
>
> **Q5. Compare with T2I-R1**
>
> We further benchmarked Vinci against T2I-R1 on both **Geneval** and **T2I-CompBench**. Vinci substantially outperforms T2I-R1 across all metrics. Notably, T2I-R1 is built upon Janus-Pro-7B—a stronger backbone than Emu3—underscoring the effectiveness of our approach.
>
> **Table 4. Geneval**
>
> | Model   | Single Obj | Two Obj | Counting | Colors | Position | Color Attr. | Overall |
> |---------|------------|---------|----------|--------|----------|--------------|---------|
> | T2I-R1  | 0.99       | 0.77    | 0.70     | 0.87   | 0.41     | 0.30         | 0.67    |
> | Vinci   | 0.99       | 0.86    | 0.48     | 0.83   | 0.86     | 0.54         | 0.76    |
>
> **Table 5. T2I-CompBench**
>
> | Model   | Color  | Shape  | Texture | Spatial | Non-Spatial | Complex |
> |---------|--------|--------|---------|---------|--------------|---------|
> | T2I-R1  | 0.8130 | 0.5852 | 0.7243  | 0.3378  | 0.3090       | 0.3993  |
> | Vinci   | 0.8225 | 0.6062 | 0.7374  | 0.3243  | 0.3148       | 0.4081  |
>
> ---
>
> **Q6. The effectiveness of the momentum-based reward function**
>
> Due to space constraints in the rebuttal, we are unable to provide additional visualizations (they will be added in the subsequent version). Instead, we report aggregated statistics over three-round scenarios in the table below.
>
>
> | Type | \<image_0> | \<image_1> | \<image_2> | w $R_m$ (%) (232/553) | w/o $R_m$ (%) (222/553) |
> |------|-----------|-----------|-----------|------------------------|--------------------------|
> | 1    | ×         | ×         | ×         | 2.16                   | 4.50                     |
> | **2**    | **×**         | **×**         | **√**         | **28.02**                  | **18.92**                    |
> | 3    | ×         | √         | ×         | 1.72                   | 6.76                     |
> | 4    | √         | ×         | ×         | 4.74                   | 8.56                     |
> | 5    | √         | √         | ×         | 4.31                   | 7.66                     |
> | 6    | √         | ×         | √         | 3.88                   | 5.41                     |
> | **7**    | **×**         | **√**         | **√**         | **51.72**                  | **45.05**                    |
> | 8    | √         | √         | √         | 3.45                   | 3.15                     |
>
> Results show that the vast majority of triple-round generations exhibit continuous improvement **(Types 2 and 7 account for ≈ 80%)**. Furthermore, the same statistics after ablating $R_m$ drop to ≈ 63% for Types 2 and 7, which once again confirms that $R_m$ is instrumental in driving the model to iteratively refine its outputs.
>
> Moreover, we will add the reward curve in the subsequent version to demonstrate the stable improvement.

---

> ### Comment · Reviewer_QUHK · 2025-08-03
>
> Thanks for the authors' rebuttal. I found in Rebuttal Table 2. T2I-CompBench, the results of Emu3 are not provided. Since Emu3 is the base model, could you provide them to better validate the improvements?

---

> > ### Author Response · Authors · 2025-08-04
> >
> > Thank you for pointing that out. We realized we missed including the results for Emu3 in Rebuttal Table 2. We have now updated the table, and you can see that our method shows significant improvements over the base model, Emu3. Most of the performance gains come from the RL phase, rather than from the SFT stage.
> >
> >
> > **Table 2. T2I-CompBench**
> >
> > | Model         | Color  | Shape  | Texture | Spatial | Non-Spatial | Complex |
> > | ------------- | ------ | ------ | ------- | ------- | ----------- | ------- |
> > | **Gen. Only** |        |        |         |         |             |         |
> > | SD3-Medium    | 0.8132 | 0.5885 | 0.7334  | 0.3200  | 0.4084      | 0.3771  |
> > | Flux.1 dev    | 0.7407 | 0.5718 | 0.6922  | 0.2863  | 0.3127      | 0.3703  |
> > | **Unified.**  |        |        |         |         |             |         |
> > | Show-o        | 0.5600 | 0.4100 | 0.4600  | 0.2000  | 0.3000      | 0.2900  |
> > | Janus-Pro-7B  | 0.6359 | 0.3528 | 0.4936  | 0.2061  | 0.3085      | 0.3559  |
> > |*Emu3* | *0.6107* | *0.4734* | *0.6178* | *0.2172*  | *0.2804* | *0.3740* |
> > | T2I-R1        | 0.8130 | 0.5852 | 0.7243  | 0.3378  | 0.3090      | 0.3993  |
> > | Vinci-sft     | 0.6274 | 0.4729 | 0.6189  | 0.2292  | 0.2711      | 0.3752  |
> > | Vinci         | 0.8225 | 0.6062 | 0.7374  | 0.3243  | 0.3148      | 0.4081  |
> >
> >
> >
> > If you have any further questions or concerns, please feel free to let us know.

---

> > > ### Comment · Reviewer_QUHK · 2025-08-07
> > >
> > > Thanks for the author's new response. Most of my concerns are addressed and I have raised my score.

---

> > > > ### Author Response · Authors · 2025-08-08
> > > >
> > > > Thank you for your thoughtful review of our rebuttal and for reconsidering your evaluation. We appreciate your constructive feedback

---

### Official Review · Reviewer_4ED1 · 2025-07-03

**Clarity:** 4
**Significance:** 3
**Originality:** 2
**Rating:** 4
**Confidence:** 4

**Summary:**

Vinci is a framework for training a multimodal, reasoning style model for generating images through iterative refinement of the image in the reasoning process. The paper first generates a cold-start dataset of reasoning chains and then use reinforcement learning to optimize according to a number of reward functions based on object detection models, or other suite if image quality evaluators. The authors finally introduce a process-based reward function based on the Adam optimizer. Vinci demonstrates strong performance across a variety of tasks.

**Questions:**

- What is the significant of having R_i and then momentum reward function? Doesn't the moment reward function also encompass R_i? I find this was the interesting ablation that was missing.
- Why does table 2 compare against the base model instead of the SFT'd model? It is unclear the benefit from SFT vs RL here.
- What is the benefit of Vinci over, say, using a fixed image generation model and doing a similar method but to rewrite the prompt instead of training the image generator.
- In the 3 generation scenario, do we actually see improvement across steps? I think this point is important to verify.

**Ethical Concerns:**

["NO or VERY MINOR ethics concerns only"]

**Final Justification:**

The authors address my concerns and the other reviewers. Thus I think this paper deserves a borderline accept.

My reasoning for not deserving a higher score is that the techniques used are the same as those in LLMs, so there is not too much innovation in terms of transferrable techniques that can help other domains, which I expect for something with a higher score.

**Limitations:**

yes

**Paper Formatting Concerns:**

The paper title and open-review submission both refer to the algorithm as Vinci, but the main text of the paper uses the word, Vinic. Some cohesion would be appreciated.

**Quality:**

3

**Strengths And Weaknesses:**

**Strengths**
- This paper is well written and easy to understand.
- Aligning models in image generation to match prompt and user intent is a significant problem that Vinci addresses.
- Vinci shows strong results on most tasks, performing well compared to leading closed source image generation models.

**Weaknesses**
- The CoT cold-start generation seems sub-optimal. Dataset collection has significant room for error after relying on the concatenation of so many models calls. Further, the authors have the model output a score from MLLM, which could be less grounded than head to head comparison, a common method for LLM as a judge comparison RLHF. Absent of the authors releasing the dataset, reproducibility may be difficult.
- Scalability of the method is limited as reasoning chains for these models can become very long due to image tokens.

---

> ### Author Rebuttal · Authors · 2025-07-31
>
> Thank you for your valuable feedback. We address each of your points individually below.
>
>
>
> **Q1. About the cold-start**
>
> We disagree with the concern that cascading multiple models necessarily amplifies error. The staged pipeline is intentionally designed to constrain error, not propagate it. By decomposing data collection into sequentially verifiable sub-tasks, we can interleave human inspection at each checkpoint, yielding lower cumulative error than any single end-to-end model that lacks such intervention points.  Moreover, the cold-start phase is scoped to *format standardization*—i.e., ensuring variable-length, interleaved text–image outputs conform to a strict schema—rather than boosting raw generation quality. Improvements in generation fidelity are eserved for the subsequent reinforcement-learning phase, whose quantitative gains are reported in *Q4*.  Due to space constraints we cannot include external links in this rebuttal, but the full dataset and codebase will be released concurrently with the camera-ready version to enable reproducibility and future extensions.
>
> ---
>
> **Q2. Scalability of the method**
>
> Thanks to the rapid progress in the community, many multimodal large language model now support context windows of up to 128k tokens, enabling the generation of substantially longer multimodal sequences [1, 2]. Complementary efforts have simultaneously compressed visual representations—e.g., Anole reduces image tokens to only 1024. We argue, however, that the current bottleneck is not the length of the reasoning chain, but rather in how to achieve better reasoning during the image generation process.
> This observation motivates the core contribution of our work: a novel end-to-end framework for interleaved text–image chain-of-thought reasoning.
>
>
> >[1] Thinking with Generated Images
>
> >[2] SAM-R1: Leveraging SAM for Reward Feedback in Multimodal Segmentation via Reinforcement Learning
>
> ---
>
> **Q3. About the $R_i$ and the momentum reward function**
>
> Thank you for the valuable suggestion. $R_i$ is employed to assess the quality of a single generated image, whereas $R_m$—the momentum-based reward function—evaluates the quality of an entire image sequence by explicitly measuring whether the model makes consistent improvements across successive generation rounds. Following your recommendation, we have added a new ablation study (see the table below).
>
> | Model                            | Single Obj | Two Obj | Counting | Colors | Position | Color Attr. | Overall |
> |----------------------------------|------------|---------|----------|--------|----------|--------------|---------|
> | Emu3-Gen                         | 0.98       | 0.71    | 0.34     | 0.81   | 0.17     | 0.21         | 0.54    |
> | w. $R_t$                         | 0.98       | 0.76    | 0.43     | 0.81   | 0.53     | 0.28         | 0.63    |
> | w. $R_i$                         | 0.96       | 0.73    | 0.36     | 0.75   | 0.72     | 0.44         | 0.66    |
> | w. $R_m$                         | 0.97       | 0.74    | 0.32     | 0.84   | 0.31     | 0.24         | 0.57    |
> | w. $R_i$ & $R_t$                 | 0.99       | 0.84    | 0.44     | 0.79   | 0.75     | 0.46         | 0.71    |
> | w. $R_i$ & $R_m$                 | 0.99       | 0.75    | 0.39     | 0.75   | 0.76     | 0.46         | 0.68    |
> | w. $R_t$ & $R_m$                 | 0.99       | 0.77    | 0.44     | 0.82   | 0.59     | 0.33         | 0.66    |
> | w. $R_i$ & $R_t$ & $R_m$         | 0.99       | 0.86    | 0.48     | 0.83   | 0.86     | 0.54         | 0.76    |
>
> The quantitative results of this ablation, together with those reported in **Q6**, confirm that $R_m$ enables the model to perform effective self-correction toward higher-quality outputs.
>
> ---
>
> **Q4. The results of SFT's model**
>
> Thank you for your suggestion. We have incorporated the results of SFT's model as shown in the table below.
>
> | Model              | Single Obj | Two Obj | Counting | Colors | Position | Color Attr. | Overall |
> |--------------------|------------|---------|----------|--------|----------|--------------|---------|
> | Emu3-Gen           | 0.98       | 0.71    | 0.34     | 0.81   | 0.17     | 0.21         | 0.54    |
> | Emu3-Gen (SFT)     | 0.98       | 0.74    | 0.39     | 0.81   | 0.50     | 0.24         | 0.61    |
> | Vinci              | 0.99       | 0.86    | 0.48     | 0.83   | 0.86     | 0.54         | 0.76    |
>
> Thank you for the insightful suggestion. We have accordingly supplemented Supervised Fine-Tuning (SFT) results, presented in the table below. While SFT yields a modest improvement (+7), the dominant gain (+15) emerges from the reinforcement-learning phase. This outcome substantiates that RL is significantly more effective than SFT in transferring the model’s visual-understanding capabilities into the image-generation process.
>
> ---
>
> **Q5. Compare with the rewrite-based methods**
>
> We appreciate the suggestion and have now included a comparative experiment against prompt-rewriting baselines; the results, shown in the table below, demonstrate that our approach consistently outperforms existing rewriting methods.
> | Model         | Single Obj | Two Obj | Counting | Colors | Position | Color Attr. | Overall |
> |---------------|------------|---------|----------|--------|----------|--------------|---------|
> | Emu3+rewrite   | 0.99       | 0.81    | 0.42     | 0.80   | 0.49     | 0.45         | 0.66    |
> | GOT [1]       | 0.99       | 0.69    | 0.67     | 0.85   | 0.34     | 0.27         | 0.64    |
> | T2i-R1 [2]    | 0.99       | 0.72    | 0.60     | 0.87   | 0.51     | 0.35         | 0.67    |
> | Vinci         | 0.99       | 0.86    | 0.48     | 0.83   | 0.86     | 0.54         | 0.76    |
>
> Moreover, our interleaved text–image chain-of-thought framework **eliminates the need for an extra rewrite model by leveraging self-reflection**, thereby reducing both computational cost and latency while delivering a superior user experience.
>
> >[1] GoT: Unleashing Reasoning Capability of Multimodal Large Language Model for Visual Generation and Editing
>
> >[2] T2I-R1: Reinforcing Image Generation with Collaborative Semantic-level and Token-level CoT
>
> ---
>
> **Q6. The improvement across steps**
>
> Due to space constraints in the rebuttal, we are unable to provide additional visualizations (they will be added in the subsequent version). Instead, we report aggregated statistics over three-round scenarios in the table below.
>
> | Type | \<image_0> | \<image_1> | \<image_2> | w $R_m$ (%) (232/553) | w/o $R_m$ (%) (222/553) |
> |------|-----------|-----------|-----------|------------------------|--------------------------|
> | 1    | ×         | ×         | ×         | 2.16                   | 4.50                     |
> | **2**    | **×**         | **×**         | **√**         | **28.02**                  | **18.92**                    |
> | 3    | ×         | √         | ×         | 1.72                   | 6.76                     |
> | 4    | √         | ×         | ×         | 4.74                   | 8.56                     |
> | 5    | √         | √         | ×         | 4.31                   | 7.66                     |
> | 6    | √         | ×         | √         | 3.88                   | 5.41                     |
> | **7**    | **×**         | **√**         | **√**         | **51.72**                  | **45.05**                    |
> | 8    | √         | √         | √         | 3.45                   | 3.15                     |
>
> Results show that the vast majority of triple-round generations exhibit continuous improvement **(Type 2 and 7 account for ≈ 80%)**. Furthermore, the same statistics after ablating $R_m$ drop to ≈ 63% for Type 2 and 7, which once again confirms that $R_m$ is instrumental in driving the model to iteratively refine its outputs.
>
> Moreover, we will add the reward curve in the subsequent version to demonstrate the stable improvement.

---

> > ### Comment · Reviewer_4ED1 · 2025-08-04
> >
> > Thank you for the rebuttal, This clarifies most of my questions. However, I am still concerned about improvement across steps. In other words, is image_0 on average scoring lower than image_1 and the same for image_2? I am not sure if I am reading the table right, but I don't understand its relation exactly to my question?

---

> > > ### Author Response · Authors · 2025-08-05
> > >
> > > Thank you for your feedback. To clarify, we understand that your question pertains to whether the generated images improve over the three iterations of scene generation. If so, the table reported in Q6 displays all possible outcomes for the three rounds of generation, where image_0/1/2 represents whether the images generated in each round are correct. For instance, Type 2 indicates that the first and second rounds generated errors, but the third round produced the correct result; Type 7 means that the first round generated an error, but the second and third rounds were correct. This table shows that, in most cases, the model is able to self-correct during the subsequent iterations (i.e., initially generating incorrect images but later correcting them).
> > >
> > > We do, however, agree with your suggestion that directly reporting the quality scores of the images would provide a clearer picture of the improvement across steps. The experiment in Q6 may not have been as intuitive, as we only counted the binary correctness of the generated images, without considering the actual scores. In response to your suggestion, **we have now included the average CLIP score for the images generated in each round**, as shown in the table below.
> > >
> > >
> > > | Model      | \<image_0\> | \<image_1\> | \<image_2\> |
> > > |------------|-----------|-----------|-----------|
> > > | w/o. $R_m$   | 0.296     | 0.314     | 0.332     |
> > > | w. $R_m$     | 0.307     | 0.338     | 0.351     |
> > >
> > >
> > > The results indicate that the quality of the images continues to improve across the three rounds of generation. Visual examples of this can be found in Figures 1(c) and 4 of the paper. In the next version, we will incorporate the experimental results outlined above, along with additional visual examples.
> > >
> > > If you have any further concerns or questions, please do not hesitate to contact us.

---

> > > > ### Comment · Reviewer_4ED1 · 2025-08-07
> > > >
> > > > I thank the authors for their response.
> > > >
> > > > This new experiment makes more sense and shows that there is indeed improvement happening between rounds. Further the authors have addressed my concerns.
> > > >
> > > > For this reason, I maintain my score in the affirmative.

---

> > > > > ### Author Response · Authors · 2025-08-08
> > > > >
> > > > > We are delighted that our response has effectively addressed your concerns. We sincerely appreciate your acknowledgment of our work and the time you have dedicated to reviewing it.

---

### Note · Authors · 2025-08-12

Dear Area Chair and Reviewers,

As for now, the paper has received 4/5 positive reviews. Reviewers `4ED1`, `QUHK`, `5wxe`, and `CRiu` have clearly stated that their concerns have been effectively addressed, and they all unanimously provided positive acceptance. Reviewer `2Qyn` did not participate in the discussion or update the final score, but we have responded to their concerns point by point.

---
## Commonly Acknowledged Contributions:

1. **Conceptual Innovation:**  First work to introduce the Interleaved MCoT within a single generative model `[ALL Reviews]`

2. **Reward Mechanism Design:**  The proposed momentum-based reward function cleverly integrates short-term generation quality with long-term improvement stability `[ALL Reviews]`

3. **Significant Performance Improvement:**  A +22% performance improvement `[ALL Reviews]`

---
## Key Verifications and Supplementary Work in the Rebuttal:

1. **Validation of Stable Multi-Round Generation Improvement:**
   CLIP scores confirmed that the quality of image_0 to image_2 steadily improved.

2. **Cross-Dataset Validity Verification:**
   The Vinci model performed excellently on the newly introduced *DPG-Bench* and *T2I-CompBench* benchmarks.

3. **Multidimensional Baseline Comparisons:**
   - Compared to the SFT model, reinforcement learning contributed the majority of the performance gain (*+15% vs +7% for SFT*).
   - Outperformed existing CoT+RL models such as *T2I-R1*, even though T2I-R1 is based on the more powerful Janus-Pro-7B backbone.
   - Surpassed optimization strategies like Best-of-N and prompt-rewrite, and achieved this without requiring additional models.
   - Outperformed RL-based T2I models like Stable Diffusion and FLUX, due to Vinci's self-reflection mechanism.

4. **Generalization Capability Verification:**
   For unseen tasks (such as the color_attr attribute not included in training), Vinci achieved a minimal gap in performance from its pre-trained results (50% vs 54%), demonstrating its effective adaptability to new scenarios.

5. **Additional Supporting Evidence:**
   Supplementary human evaluation results confirm that Vinci outperforms all baseline models in subjective assessments. The theoretical convergence analysis of the momentum-based reward function and statistics on computational costs (FLOPs/memory) were also provided.

---
Thanks to all the reviewers and their valuable time and effort dedicated to our work

Best,

Authors

---

### Decision · Program_Chairs · 2025-09-17

**Decision:**

Accept (poster)

**Comment:**

This paper introduces Vinci, a framework for text-to-image generation that integrates Multimodal Chain-of-Thought (MCOT) reasoning with RL. The core is the use of a unified autoregressive model that interleaves image generation and understanding, allowing for a "deep thinking" process of iterative refinement. The authors propose a momentum-based reward function to stabilize the RL training process and demonstrate a +22% improvement over the base model on the GenEval benchmark. The work is relevant to the NeurIPS community as it addresses the critical challenge of enhancing the reasoning and self-correction capabilities of generative models.

The paper received four positive-leaning reviews (5, 4, 4, 4) and one borderline negative review (3). After the author rebuttal, all four of the initially positive-leaning reviewers confirmed their ratings, and one raised their score. The initially negative-leaning reviewer did not engage beyond the acknowledgement. Furthermore, the reviewers were generally confident in their assessments (4, 4, 3, 3, 4).

Reviewers acknowledged the conceptual novelty of integrating MCOT within a single generative model and the innovative momentum-based reward function. The significant improvement on benchmark datasets is also highlighted as a major strength.

Initially, several reviewers raised concerns about the limited evaluation scope (only GenEval), the lack of comparison with certain state-of-the-art models, and the reliance on synthetic data for the cold-start phase. There was the initial "borderline reject" rating from one reviewer, who was concerned about the evaluation scope and potential for reward hacking. However, the other reviewers, while acknowledging these as limitations, felt the novelty and strong results outweighed these concerns. The authors' extensive rebuttal, which included new experiments on additional benchmarks and a human evaluation, successfully addressed most of these concerns for the other reviewers.

The proposed momentum-based reward function is well-motivated - the authors also provided a theoretical analysis of its stability and convergence properties in the rebuttal, which was well-received.

The initial experiments on GenEval were strong, and the authors significantly strengthened their empirical validation during the rebuttal by adding results on DPG-Bench and T2I-CompBench, as well as human evaluation data. They also included important ablation studies and comparisons with additional baselines as requested by the reviewers. Also, authors report a practical trade-off: runtime and memory as an overhead of multi-round “deep thinking”. While the code and data were not available at the time of review, the authors have committed to releasing them with the camera-ready version. The paper provides a good level of implementation detail, which was further clarified in the rebuttal.

The authors initially omitted comparisons to some relevant RL-based text-to-image models but added these during the rebuttal, demonstrating the advantages of their approach.

This work has the potential to have a nice impact on the sub-area of multimodal AI. By demonstrating a successful method for enabling a generative model to "think" and iteratively refine its own output, this paper opens up new avenues for research into more controllable and reliable generative models. The proposed techniques could be influential for the development of future unified multimodal systems. The proposed method is also sound and supported by strong empirical evidence, which was substantially strengthened during the author-reviewer discussion period. The work is a valuable contribution to the field of multimodal AI.  The authors must incorporate the additional experiments, analyses, and implementation details from the rebuttal into the final paper.